# Conformational dynamics of the human serotonin transporter during substrate and drug binding

Ingvar R. Möller [1,2], Marika Slivacka[1,2], Anne Kathrine Nielsen [2], Søren G.F. Rasmussen[3], Ulrik Gether [3], Claus J. Loland [2] & Kasper D. Rand [1]

The serotonin transporter (SERT), a member of the neurotransmitter:sodium symporter family, is responsible for termination of serotonergic signaling by re-uptake of serotonin (5-HT) into the presynaptic neuron. Its key role in synaptic transmission makes it a major drug target, e.g. for the treatment of depression, anxiety and post-traumatic stress. Here, we apply hydrogen-deuterium exchange mass spectrometry to probe the conformational dynamics of human SERT in the absence and presence of known substrates and targeted drugs. Our results reveal significant changes in dynamics in regions TM1, EL3, EL4, and TM12 upon binding co-transported ions ($Na^+/K^+$) and ligand-mediated changes in TM1, EL3 and EL4 upon binding 5-HT, the drugs S-citalopram, cocaine and ibogaine. Our results provide a comprehensive direct view of the conformational response of SERT upon binding both biologically relevant substrate/ions and ligands of pharmaceutical interest, thus advancing our understanding of the structure-function relationship in SERT.

[1] Protein Analysis Group, Department of Pharmacy, University of Copenhagen, Copenhagen 2100, Denmark. [2] Laboratory for Membrane Protein Dynamics, Department of Neuroscience, University of Copenhagen, Copenhagen 2200, Denmark. [3] Department of Neuroscience, University of Copenhagen, Copenhagen 2200, Denmark. Correspondence and requests for materials should be addressed to C.J.L. (email: cllo@sund.ku.dk) or to K.D.R. (email: kasper.rand@sund.ku.dk)

Serotonin (5-HT) as a neurotransmitter is involved in many basal brain functions such as mood, sleep, and appetite. The intensity of 5-HT transmission is tightly regulated by fast removal of released 5-HT by the serotonin transporter (SERT), which is embedded in the presynaptic membrane. The transport by SERT is a $Na^+$-coupled secondary active process, meaning that it harnesses the energy in the $Na^+$-gradient across cell membranes, to drive the transport of 5-HT against its concentration gradient. For every molecule 5-HT, one $Na^+$ and one $Cl^-$ ion is co-transported[1,2]. A concomitant counter transport of $K^+$ renders the transport process electroneutral[3].

The overarching importance of 5-HT in basal physiological functions makes SERT a natural drug target in the treatment of multiple mental diseases. Accordingly, the selective serotonin reuptake inhibitors (SSRIs), such as S-citalopram, sertraline, and paroxetine, are currently in use for the treatment of, e.g., depression, anxiety, obsessive–compulsive disorders, and post-traumatic stress disorder. The tricyclic antidepressants, e.g., imipramine and amitriptyline also target SERT. In addition, SERT inhibition contributes to the action of psychotropic and illicit drugs such as MDMA (ecstasy), ibogaine, cocaine, and amphetamine[4].

SERT is a member of the neurotransmitter:sodium symporter (NSS) family that also includes the transporters for the neurotransmitters dopamine, norepinephrine, glycine, and γ-aminobutyric acid. The family moreover encompasses multiple other transporters for amino acids and metabolites[5]. The non-polar amino acid transporter, LeuT, from *Aquifex aeolicus* was the first NSS member for which a crystal structure was solved[6]. The structure revealed a protein with 12 transmembrane (TM) spanning domains with a pseudo twofold symmetry between TM1–5 and 6–10, now known as the LeuT-fold. A binding site (S1 site) for the substrate leucine was found to be located to the center of the protein flanked by TM1, 3, 6, and 8. Two sodium ions were bound in the structure; one (Na1 site) in direct contact with the bound leucine and the other (Na2 site) in close proximity[6].

NSS proteins are thought to function by an alternating access mechanism by which the central binding site is alternatingly accessible to either the extracellular or intracellular environment. Accordingly, the transporter exists in a dynamic equilibrium between different conformations, which can be biased toward a specific conformation by the binding of ions, substrates, or specific inhibitors. X-ray structures have been solved of LeuT in three distinct conformations supporting the alternating access mechanism: a $Na^+$-bound outward open, a $Na^+$ and leucine-bound outward occluded, and an inward open apo-form[6,7]. While providing detailed structural insight, these structures represent static "snapshots" of a dynamic protein.

Structural dynamics of NSS proteins have been investigated extensively in LeuT with several biophysical methods such as electron spin resonance[8,9], fluorescent- and fluorescence resonance energy transfer (FRET) spectroscopy studies[10–14] and hydrogen–deuterium exchange (HDX) mass spectrometry (MS)[15,16]. They show that the binding of $Na^+$ and leucine promotes an opening of the outer gate, destabilizing the extracellular domains and increases their solvent accessibility toward the binding site[8–14,16]. Conversely, the intracellular domains are stabilized by this action. We have previously used HDX-MS on LeuT to make the intriguing observation that the binding of solutes promotes slow cooperative structural fluctuations of the intracellular part of TM1, 5, and 7 as well as within the binding site region of TM6[15]. An helical unwinding of TM5 has also been observed in other bacterial members of the NSS family[17]. It has been proposed that the function is a priming of LeuT for subsequent intracellular release of $Na^+$ and substrate through the

release of an N-terminal interaction[18]. A role of $K^+$ has been shown for SERT function, that is $K^+$ binds to the inward-facing transporter and is transported out of the cell, rendering the transport charge-neutral[19] and speeding up the return-step of the transport cycle[20]. Interestingly, recent findings suggest a role of $K^+$ also in LeuT. For this transporter, it was found that $K^+$ inhibits $Na^+$ and substrate binding, and biases the transporter toward an inward facing conformation, thereby stabilizing the extracellular domains and destabilizing the intracellular face[13,15].

The solved structures of both hSERT[21] and dDAT[22] have shown an overall fold very similar to LeuT, suggesting a structural and mechanistic conservation within the NSS family of proteins. The hSERT crystal structure revealed an outward-open conformation with S-citalopram or paroxetine bound in the S1 site. The binding sites for two $Na^+$ and a $Cl^-$ ion are also found in the structure. All three ions are located similar to the positions found in dDAT[21,22]. In addition to the central binding site, an additional S-citalopram molecule is bound within an extracellular vestibule (the S2 site), a cone-shaped water-filled cavity which provides the entry pathway for the S1-bound molecule as well as $Na^+$ and $Cl^-$, confirming previous findings[23]. However, in spite of the apparent structural similarity between LeuT and SERT, and several MD simulations of SERT[24–26], our knowledge from experimental data on the conformational dynamics of SERT is incomplete and based on indirect inferences from biochemical and electrophysiological studies[27–31] often using SERT mutants. A greater insight into these aspects would be of utmost importance for a better understanding of 5-HT homeostasis and how this can be affected by current and future medications.

HDX-MS provides an attractive method to study the conformational dynamics of membrane proteins[32–34] and has been used to study transporters including the effect of substrates on LeuT[15,16], the effect of interactions between phospholipids and well-characterized model transporters from *E. coli*[35], and the effect of consensus mutagenesis for thermostabilization of the human glutamate transporter[36]. By this technique, the exchange of hydrogen to deuterium of backbone amides in a protein is measured by use of MS. Amide HDX is primarily determined by the presence and strength of local hydrogen bonding. Changes in the rate of HDX is, thus, closely correlated to conformational properties and dynamics of a protein[37,38]. An advantage with HDX-MS is it allows for probing the conformational dynamics of the entire protein backbone in a non-perturbing manner and readily facilitates comparative analyses between different protein conditions[39–42].

Here, we apply HDX-MS to probe the conformational dynamics of hSERT induced by the addition of ions ($Na^+$ or $K^+$), substrate (5-HT) or inhibitors (S-citalopram, cocaine, and ibogaine). By measuring changes in deuterium uptake as a function of ion- and ligand-bound states, we map ligand-induced differences in structural stabilities within specific hSERT domains. We find that $K^+$ and $Na^+$ bound states show significantly different HDX in regions which have been implicated in the transition from outward to inward facing conformations but also in regions with hitherto unknown roles. The binding of known substrate (5-HT) and drug ligands (S-citalopram, cocaine, and ibogaine) has diverse conformational impacts on hSERT, primarily observed through their effect on segments spanning TM1a and EL3, thus providing molecular insights into the substrate transport mechanism and the binding of drugs.

## Results

**Purification of human SERT**. To enable HDX-MS experiments, hSERT was expressed in Expi293F cells using the BacMam procedure[43]. The cells were harvested and lysed by sonication. The

membranes were collected by centrifugation and solubilized in a buffer containing 20 mM dodecyl maltoside (DDM). DDM-solubilized hSERT was purified by affinity chromatography and eluted into a stabilizing buffer containing DDM (1 mM) and a mixture of lipids (POPC:POPE:POPG; 1:1:1 ratio). Sodium dodecyl sulphate polyacrylamide gel electrophoresis analysis of the purification indicated around 20–25% purity and the absence of SERT proteolysis products in the hSERT sample used for HDX-MS was confirmed by western blotting analysis (Supplementary Fig. 1). The activity of purified DDM-solubilized hSERT was verified by a saturation binding assay using [³H] imipramine (Supplementary Fig. 1). The $K_d$ was measured to be $7.5 \pm 0.7$ nM (mean $\pm$ SE, $n = 3$), which correlated with previous reports[1,44]. Additional analyses of the purity and verification of stability of purified hSERT can be found in Supplementary Figs. 1 and 2.

**Conformational dynamics of hSERT in solution**. Extensive HDX-MS optimization using on-line rhizopuspepsin digestion[45] allowed the identification of 62 peptides covering 69.1% of the protein sequence that were used to measure the HDX of local regions of hSERT across all states examined (Supplementary Results and Supplementary Fig. 3 for further details).

According to the crystal structure[21], the 12 TM helices make up approximately 52% of hSERT protein sequence and at least 12% form helical structural motifs in loops and terminal areas. Helices are primarily stabilized by hydrogen bond networks and the HDX of these regions would be expected to be reduced compared to that of unstructured parts of the transporter. We thus used the HDX of hSERT measured at the earliest time point (0.25 min) in a Na⁺-containing buffer to detect the presence of hydrogen-bonded structure elements of hSERT in solution (Fig. 1)[38]. As Na⁺ is required for transport, Na⁺ was used as the reference state for investigating the conformation of hSERT and furthermore investigating the relative conformational impact of ligands.

In most instances, the deuterium uptake of regions shown to comprise TMs in the crystal structure followed a pattern of having slow HDX corresponding to the formation of TMs in the solution phase. However, parts of TM2, TM3, TM6, and TM10, displayed faster HDX than expected, resulting in over 40% of the backbone amides already being labeled with deuterium after 0.25 min, compared to a maximum labeled control sample. In the case of TM2, the peptides included both the TM region and an unstructured segment where the latter could explain the observed elevated HDX. However, the extracellular part of TM3 (peptide number 26), intracellular parts of TM6 (peptide number 37), and the extracellular part of TM10 (peptide number 47) did not include unstructured segments. The flexibility could make them particularly responsive to conformational transitions.

We note that the observed flexibility of parts of TM3, TM6, and TM10 could be due to local structural perturbations caused by the nonnative solubilization conditions employed (DDM detergent micelle). However, the observed full binding activity of detergent-solubilized SERT used for HDX-MS makes this unlikely and furthermore the X-ray crystal structure of SERT reveals these helices to be partially solvated, which provide an alternate explanation for their elevated HDX To address whether HDX in some SERT domains could be influenced by partly unfolding, we performed HDX-MS experiments on purified SERT batches after sample pretreatment steps that involved storage at room temperature for 1 h. All samples yielded similar HDX profiles (Supplementary Fig. 2). Such locally resolved measurements are very sensitive to detect even modest local changes in SERT conformation due to partial unfolding or aggregation during the time-frame of the HDX experiment.

**The impact of K⁺ on the conformational dynamics of hSERT**. Unlike any other mammalian NSS family member, the transport cycle of hSERT has been reported to possess a counter-transport of K⁺[4]. To investigate the effect of K⁺ on hSERT conformation, we performed HDX of hSERT in a K⁺-containing buffer and compared to hSERT in Na⁺-containing buffer (reference state, Figs. 1 and 2).

K⁺ induced significant changes in HDX (and hence conformational dynamics) in multiple regions of hSERT relative to the Na⁺-bound state (Fig. 2). Decreased HDX was observed in a segment spanning the upper part of TM6a (residues 319–329) and segments within the two helices and interconnecting loop in EL4 (residues 397–406). Increased HDX was observed on the intracellular face in residues located in the N-terminus (residues 70–83), TM1a (residues 84–95), EL3 (residues 296–305 and 312–320), along with the intracellular parts of TM12b (residues 595–598). Together, these data are consistent with a stabilization or closing of this continuous extracellular region of hSERT and a coordinated destabilization or opening of the intracellular segments when converting to a K⁺-containing buffer.

The presence of K⁺ also induced unusually slow unfolding–refolding kinetics (EX1 kinetics) in TM1b (residues 99–109). Proteins commonly undergo HDX according to a regime where the breakage and formation of amide H-bonds is uncorrelated and faster than the rate of HDX, which is characterized by a single binomial isotopic distribution that gradually migrates to higher mass-to-charge (m/z) with increased deuteration of the protein (EX2 kinetics). EX2 kinetics was observed in all regions of hSERT across all states, apart from TM1b in the K⁺-bound state. EX1 kinetics reflects slow cooperative structural fluctuations where the reformation of H-bonds is slower than the rate of HDX. This causes a correlated exchange of the backbone amide hydrogens allowing HDX-MS to detect two protein populations present in solution: an already-exchanged population and a yet-to-exchange population. As the two populations have not incorporated the same amount of deuterium, their mass differs, and they can be distinguished from each other by mass analysis. For TM1b (residues 99–109), the two populations differed in HDX by six deuteriums (Supplementary Fig. 4), indicating that in the presence of K⁺, at least six backbone amide hydrogens—corresponding to almost two helical turns—undergo correlated exchange in this part of hSERT. It suggests that the dynamics of the extracellular side of TM1 is considerably slowed by the presence of K⁺. TM1b forms part of the extracellular gate and is known to be involved in substrate transport. By extension, our results suggest that the opening and closing motions of the extracellular gate is significantly modulated and slowed by the binding of K⁺ relative to the Na⁺-bound SERT.

In control experiments, the impact on the dynamics of hSERT by imidazole from the purification buffer was evaluated by measuring the HDX of hSERT in the presence and absence of imidazole, both in Na⁺ and K⁺ buffers. No substantial differences in HDX were observed for hSERT with and without imidazole in Na⁺ buffer. Intriguingly, some HDX differences were observed for hSERT in K⁺ buffer. Segments covering TM1b (residues 99–109), part of TM5 along with EL3 (residues spanning from 296 to 320), TM10 and TM12 with the C-terminally located helix showed increased HDX in the presence of imidazole (Supplementary Fig. 5). Notably, all subsequent ligand-binding experiments were performed in Na⁺ buffer.

**5-HT binding induces distinct conformational dynamics**. hSERT is responsible for the reuptake of the 5-HT from the synaptic cleft. To probe the conformational dynamics induced by

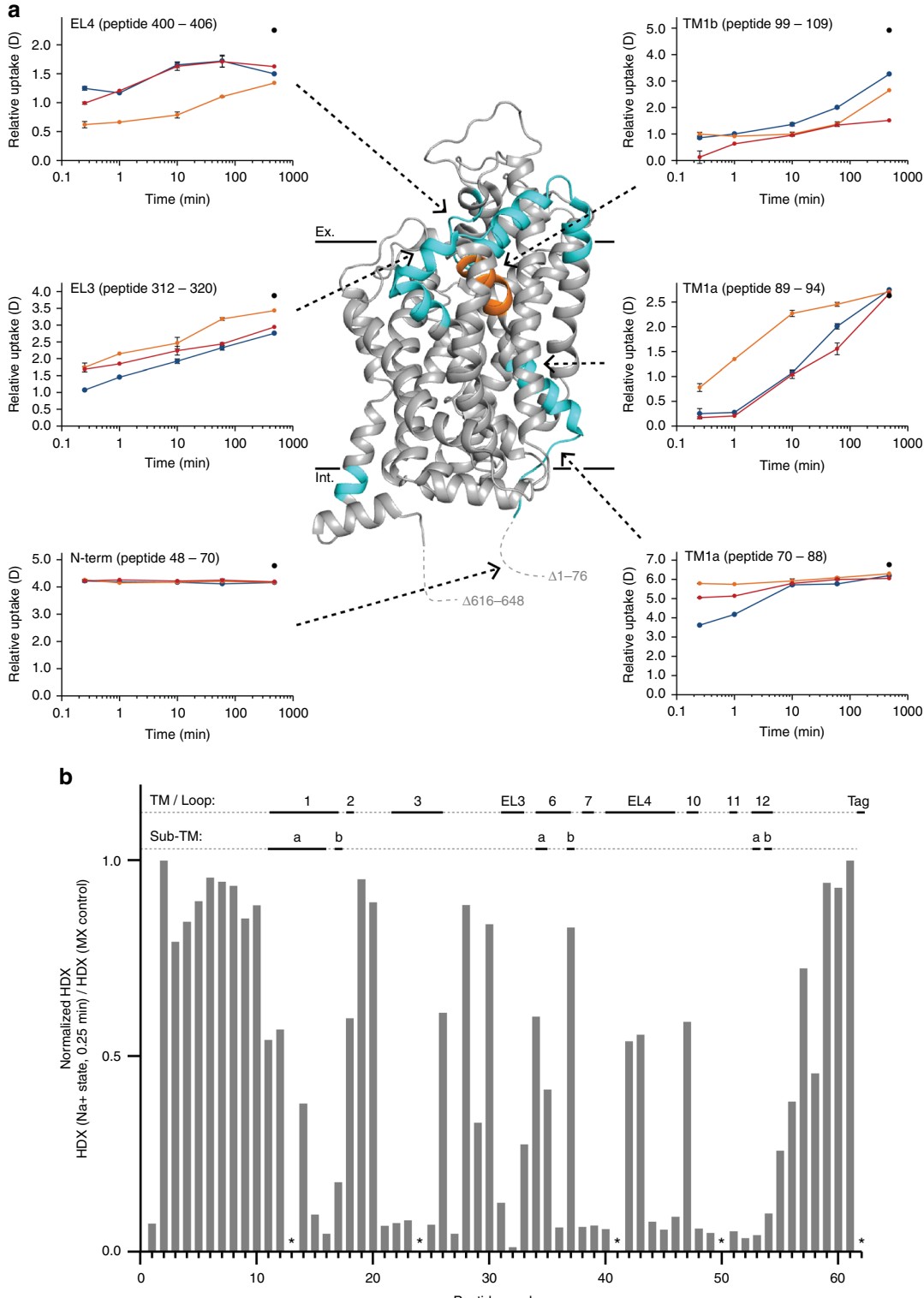

**Fig. 1** Conformational dynamics of hSERT in solution. **a** The relative deuterium uptake of representative peptides is plotted as a function of labelling time (0.25–480 min) when hSERT was in the presence of $Na^+$ (blue), $K^+$ (orange), and $Na^+$ with 5-HT (red). Error bars indicate standard deviations for the 0.25, 10, and 60 min time points ($n = 2$–4). Maximum labelled controls are indicated with a black dot. Peptides showing a difference in deuterium uptake between at least two states are coloured cyan on the crystal structure (PDB: 5i75). Orange indicates that EX1 kinetics was observed for at least one experimental state. Black solid lines at the top and bottom of the structural model show an approximation of the extracellular (Ex.) and intracellular (Int.) boundaries of the membrane, respectively. **b** The HDX of hSERT measured at the earliest time point (0.25 min) in the presence of $Na^+$, normalized to the maximum labelled control sample, and plotted as a function of peptide number (Supplementary Table 1). Structural elements are indicated along the x-axis above the plot. (*) marks peptides where information was not available. Source data for the HDX data are provided as a Source data file

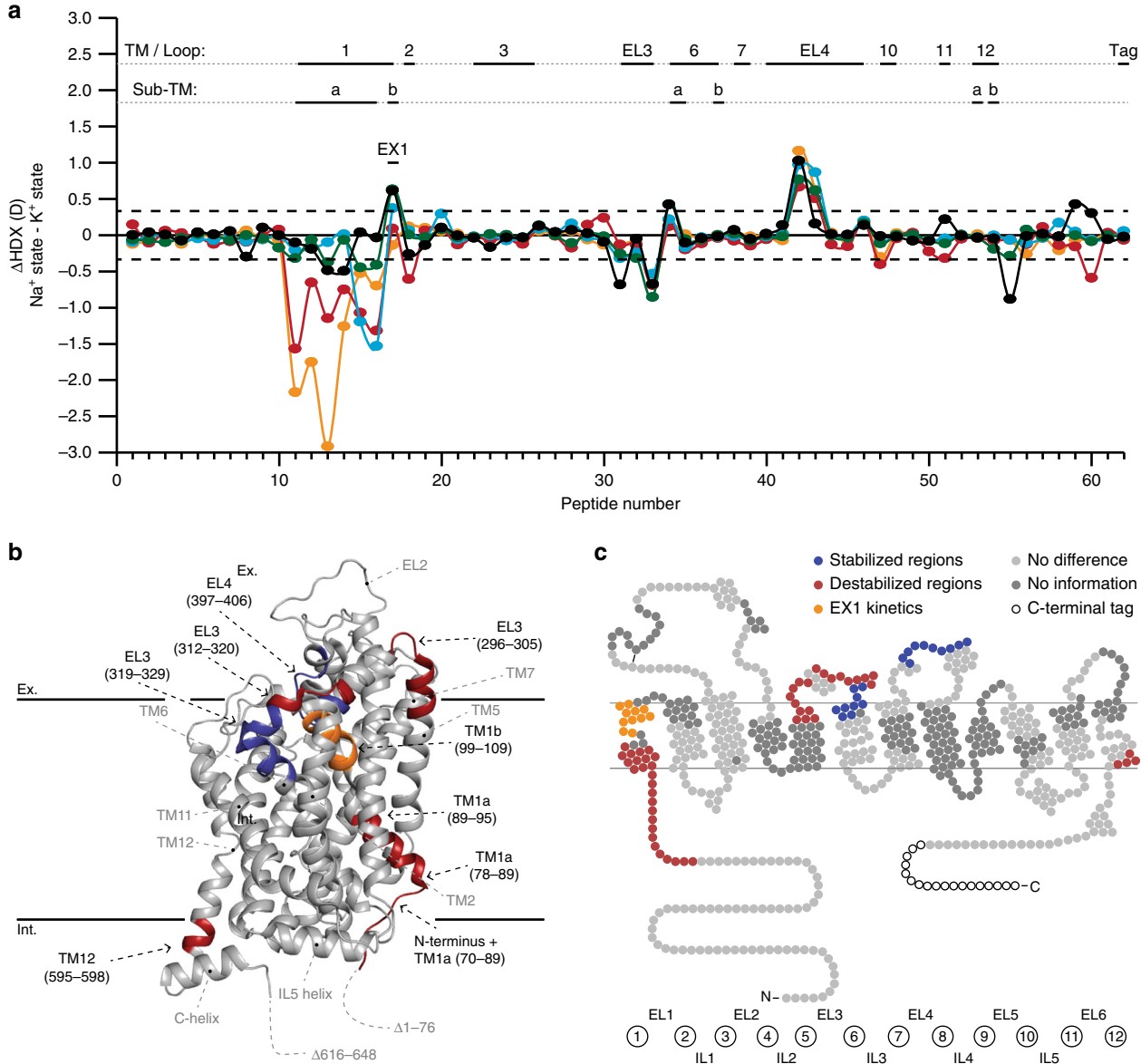

**Fig. 2** Comparison of the HDX of hSERT in the presence of Na+ and K+. **a** A difference plot illustrating the difference in HDX between the Na+ and K+ states for the 62 peptides identified from hSERT over the measured time points (orange—0.25 min; red—1 min; cyan—10 min; green—1 h; black—8 h). Positive and negative values indicate decreased or increased HDX, respectively, when hSERT is in the presence of K+ compared to Na+. The peptides are arranged according to their position from the N- to C-terminal (Supplementary Table 1). A dotted line is plotted at ±0.34D, indicating the 95% CI as a threshold for significance. Structural motifs in hSERT are indicated along the x-axis above the plot as well as the occurrence of EX1 kinetics. **b**, **c** Regions showing significant increase (red) or decrease (blue) in dynamics (i.e., HDX) in the presence of K+ compared to Na+ are plotted on the crystal structure (PDB ID: 5i75) (**b**) and a snake diagram (**c**) of hSERT. Regions coloured grey indicates insignificant difference in HDX between the two experimental states (light grey) or regions not covered by peptides (dark grey). Regions showing significant difference in HDX are indicated with black text with residue location in the structural model. Structural elements located on the front side of the crystal structure but do not show significant difference in HDX are indicated with grey text without residue information for orientation. Regions not resolved or truncated in the crystal structure are indicated with dotted lines. Black solid lines at the top and bottom of the structural model show an approximation of the extracellular (Ex.) and intracellular (Int.) boundaries of the membrane, respectively. Source data for the HDX data are provided as a Source data file

5-HT binding, we measured the HDX of hSERT in the presence of 5-HT (10 μM) and compared it to the Na+-bound state (Figs. 1 and 3).

5-HT induced significant changes in the dynamics of the protein resulting in HDX distinguishable from that observed when the transporter was in the presence of either K+ or Na+ (Fig. 3). Similar to K+, 5-HT binding increased HDX in the N-terminal tail immediately prior to TM1a (residues 70–83). This domain includes a conserved arginine residue (Arg79), which has

been proposed to form a salt bridge to Asp452 in IL4[7,46]. Its breakage likely mediates the transition towards the inward facing state. The first five residues of the TM1a helix itself (residues 84–89) were also destabilized. Increased HDX was also observed in EL3 (segments covered by peptides 306–311 and 312–320) analogue to the K+-bound state. A decreased HDX was observed for the remainder of the TM1 segment (residues spanning 89–95 + 99–109). These changes in stabilization is probably a signature for a transition toward the inward facing

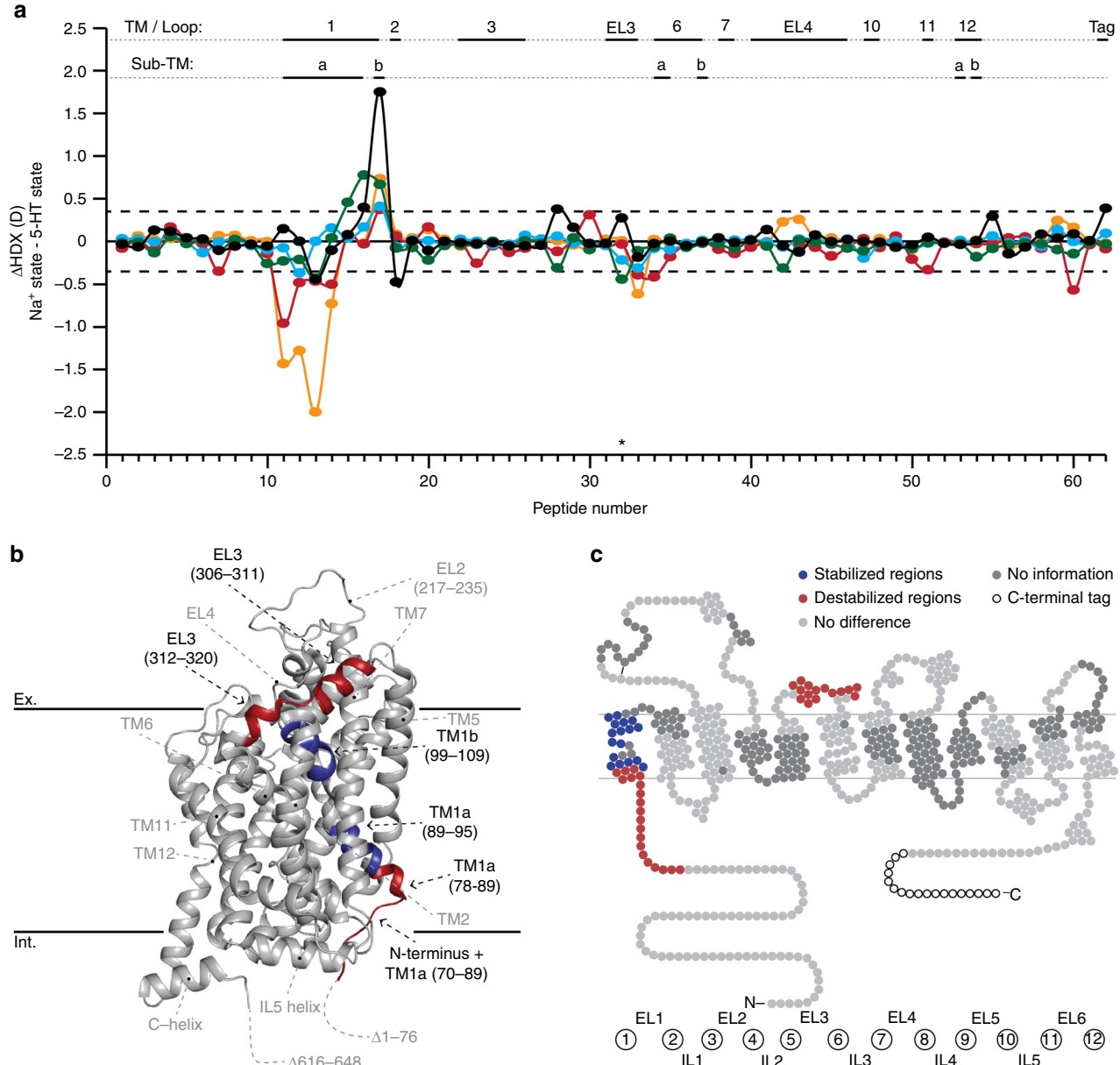

**Fig. 3** Comparison of the HDX of hSERT in the presence of Na$^+$ and 5-HT with Na$^+$. **a** A difference plot illustrating the difference in HDX between the Na$^+$ and 5-HT states for the 62 peptides identified from hSERT over the measured time points (orange—0.25 min; red—1 min; cyan—10 min; green—1 h; black—8 h). Positive and negative values indicate decreased or increased HDX, respectively, when hSERT is in the presence of 5-HT and Na$^+$ compared to Na$^+$ only. The peptides are arranged according to their position from the N- to C-terminal (Supplementary Table 1). The dotted line (±0.35$D$) indicates 95% CI as a threshold for significance. Structural motifs in hSERT are indicated along the $x$-axis above the plot. **b, c** Regions showing significant increase (red) or decrease (blue) in dynamics (i.e., HDX) in the presence of 5-HT and Na$^+$ compared to Na$^+$ alone are plotted on the crystal structure (PDB ID: 5i75) (**b**) and a snake diagram (**c**) of hSERT. Regions coloured grey indicate insignificant difference in HDX between the two experimental states (light grey) or regions not covered by peptides (dark grey). Regions showing significant difference in HDX are indicated with black text with residue location in the structural model. Structural elements located on the front side of the crystal structure but do not show significant difference in HDX, are indicated with grey text without residue information for orientation and regions not resolved or truncated in the crystal structure are indicated as dotted lines. Black solid lines at the top and bottom of the structural model show an approximation of the extracellular (Ex.) and intracellular (Int.) boundaries of the membrane, respectively. Source data for the HDX data are provided as a Source data file

state. We also observed significant differences between 5-HT and K$^+$ bound states, particularly in EL4 and the C-terminal part of TM12. It could suggest that the changes are induced by Na$^+$-binding and not further modulated by substrate. The data presented here clearly suggest a difference in the substrate-bound conformations of LeuT and hSERT, whereas LeuT is further stabilized in its Na$^+$-bound state by leucine[10,15], 5-HT binding to

hSERT both stabilizes but also destabilizes regions relative to the Na$^+$-bound form. hSERT has been proposed to favor an inward-facing conformation in the presence of 5-HT[47], and the structural stabilization observed in TM1b by 5-HT may be indicative of a closing of the extracellular gate. Our HDX data, however, does not support a complete transition to an inward-facing state similar to suggestions in previously reported MD simulations[25].

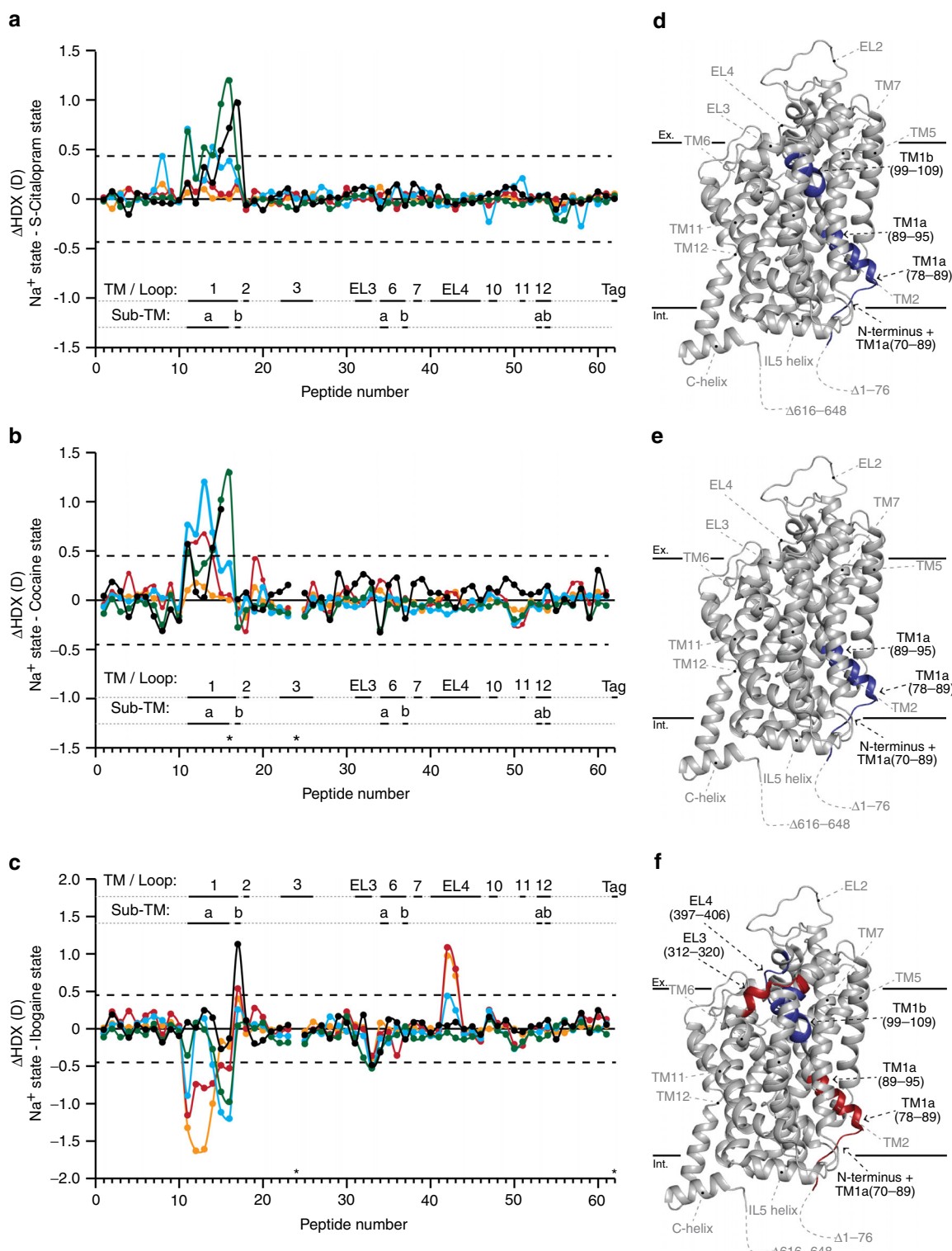

**Structural stabilization upon binding of S-citalopram.** S-citalopram is a widely used antidepressant, which inhibits transport of 5-HT by SERT through competitive inhibition[48]. To evaluate the effect of S-citalopram on the conformational dynamics of hSERT and detect regions directly or indirectly affected by ligand binding, the HDX of the transporter was performed in the presence of S-citalopram and compared to its $Na^+$-bound state.

The binding of S-citalopram led to significant structural stabilization in TM1 (Fig. 4a, d). Decreased HDX was observed

in the region immediately preceding TM1 as well as throughout the TM1 domain (residues 70–95 and 99–109, residues 96–98 were not covered by any peptides). According to the crystal structure, TM1a participates in the binding of S-citalopram through Tyr95 which forms a π–π interaction with its amine group and possibly through a hydrogen bond with its oxygen[21]. However, residues not reported to participate directly in S-citalopram binding, such as the N-terminal tail (residues 70–94) also underwent stabilization. This could possibly be due to allosteric restraints brought on by the inhibitor binding such as

**Fig. 4** Comparison of the HDX of hSERT in the presence and absence of drug inhibitors. Difference plots illustrating the difference in HDX between the $Na^+$ and inhibitor states (**a** S-citalopram; **b** cocaine; **c** ibogaine) for the 62 peptides identified from hSERT over the measured time points (orange—0.25 min; red —1 min; cyan—10 min; green—1 h; black—8 h). Positive and negative values indicate decreased or increased HDX, respectively, when hSERT is in the presence of each respective inhibitor with $Na^+$ compared to $Na^+$ only. The peptides are arranged according to their position from the N- to C-terminal (Supplementary Tables 2 and 3). A dotted line is plotted at ±0.43 (**a**) and ± 0.45 D (**b**, **c**), indicating the 95% CI as a threshold for significance. Structural motifs in hSERT are indicated along the x-axis above or below each plot. **d**–**f** Regions showing significant increase (red) or decrease (blue) in dynamics (i.e., HDX) in the presence of inhibitor (**d** S-citalopram, **e** cocaine, **f** ibogaine) with $Na^+$ compared to $Na^+$ alone are plotted on the crystal structure of hSERT (PDB ID: 5i75). Regions coloured grey indicates insignificant difference in HDX between experimental states or regions not covered by peptides. Regions showing significant difference in HDX are indicated with black text with residue location in the structural model. Structural elements located on the front side of the crystal structure but do not show significant difference in HDX, are indicated with grey text without residue information for orientation and regions not resolved or truncated in the crystal structure are indicated with dotted lines. Black solid lines at the top and bottom of the structural model show an approximation of the extracellular (Ex.) and intracellular (Int.) boundaries of the membrane, respectively. Source data for the HDX data are provided as a Source data file

the stabilization of the conserved salt bridge between Arg79 and Asp452. In our previous HDX-MS study on LeuT, the outward-facing state of LeuT in the presence of $Na^+$ showed structural destabilization in TM1b[15] which is opposite to what is observed for hSERT, here in the presence of S-citalopram. TM1b is known to be important in transporting substrate, making up a part of the extracellular gate, participating in coordination of Na1 as well as containing a salt bridge between Arg104 and Glu493[21]. Its structural stabilization could be caused by S-citalopram restricting the movement of TM1b and possibly preventing hSERT from isomerizing into the outward-facing occluded conformation.

**Binding of cocaine leads to stabilization of TM1a.** Cocaine is a substance of abuse which primarily exerts its stimulant effects through DAT inhibition[49] and increasing the synaptic concentration of dopamine[50]. However, cocaine's nonselective nature also causes it to inhibit transport of 5-HT through competitive inhibition of hSERT[51]. To gain insight into the hSERT binding conformation induced by cocaine, we performed HDX on hSERT in the presence of 2.4 μM cocaine and compared the effects to the $Na^+$-bound state.

Relative to the $Na^+$-bound state, the addition of cocaine led to stabilization of TM1a (residues 84–95) and the region immediately before the TM1 helix (residues 70–83) (Fig. 4b, e). As for S-citalopram, cocaine binding is reported to promote an outward-facing conformation of hSERT[47] through an interaction with Tyr95[52], perhaps causing a stabilization of the salt bridge between Arg79 and Asp452 on the intracellular side. However, the lack of sequence coverage in TM8-IL4-TM9 eludes the identification of a possible corresponding stabilization in IL4. Unlike S-citalopram, the binding of cocaine to hSERT did not have a significant impact on the dynamics of TM1b (residues 99–109), indicating that the hSERT binding dynamics between the two compounds is different.

**Similar conformational response in hSERT to ibogaine and $K^+$.** The phychedelic alkaloid ibogaine from the African shrub Tabernanthe iboga has been suggested as a possible treatment for drug addiction[53]. It is also an inhibitor of SERT although less potent than S-citalopram and cocaine. It has been proposed that ibogaine inhibits 5-HT transport through a noncompetitive action and possibly inducing an inward-open SERT conformation[54,55]. If this is correct, it would likely result in a HDX pattern which is markedly different from our observations for S-citalopram and cocaine. Accordingly, we performed HDX-MS on purified hSERT in the presence of ibogaine.

Relative to the buffer with $Na^+$ only, the addition of ibogaine led to a structural destabilization of the region immediately before TM1a as well as TM1a itself (residues 70–95) along with a

segment spanning EL3 (residues 312–320). Oppositely, stabilization was observed in segments spanning TM1b (residues 99–109) and parts of EL4a and EL4b (residues 397–406) (Fig. 4c, f). The observed effects in TM1 are literally opposite to our observations for cocaine and S-citalopram but similar to the effects on hSERT in $K^+$-containing buffer. Based on HDX-MS on LeuT, we have previously proposed that the stabilization of TM1b and EL4 and the destabilization of TM1a were connected with the transition towards an inward-facing conformation[15]. Here, the destabilization of TM1a and stabilization of TM1b by ibogaine could indicate an initiation towards a similar transition. Also the stabilization of EL4 supports this suggestion as the tip of EL4 in LeuT has been shown by X-ray crystallography to act as a lid for the closure of the extracellular vestibule in LeuT[7].

## Discussion

The structural models of NSS proteins, from bacteria to hSERT have been of utmost importance in studying the protein structure and possible mechanism of NSS proteins[6,7,17]. However, truncations, mutations, or deletions have been required for the crystallization of the eukaryotic NSS proteins, sometimes even rendering them incapable of transport. In other complementary studies, techniques such as electron spin resonance[8,9], fluorescence-[10] and FRET-based[11–13] strategies have been applied, but they also require modifications to the protein to conjugate the reporter probes. A major advantage of HDX-MS is that it can be performed on an intact and unmodified protein, still capable of assuming all possible functional states. We have recently reported the HDX-MS study of LeuT wild type in the presence and absence of substrate/ions, providing insights into the solution-phase conformational dynamics of this prokaryotic transporter[15]. However, it is currently unknown to what extent our knowledge of the conformational dynamics and substrate/ion-induced changes in LeuT reflect those occurring in human neurotransmitter transporters.

Here, we provide insights into the solution-phase conformational dynamics of purified hSERT in the presence of $Na^+$, $K^+$, 5-HT, and selected inhibitors. In the presence of $K^+$, structural destabilization was observed in hSERT immediately prior to TM1a, TM1a itself, and EL3 while TM1b and EL4 became stabilized. We have previously reported similar structural destabilization in LeuT, showing that the two transporters share similarities in their response to $K^+$. Compelling data suggest that $K^+$ is counter-transported relative to the inward transport of $Na^+$ and 5-HT[56], which could support a bias toward a more inward-facing structure by $K^+$. The impact of $K^+$ on hSERT revealed in this study clearly support this hypothesis and maps which regions are involved in this transition (peptides spanning residues 70–89, 89–95, 99–109, 296–305, 312–329, 397–406, and 400–406). LeuT was also shown to favor an inward-facing conformation in the

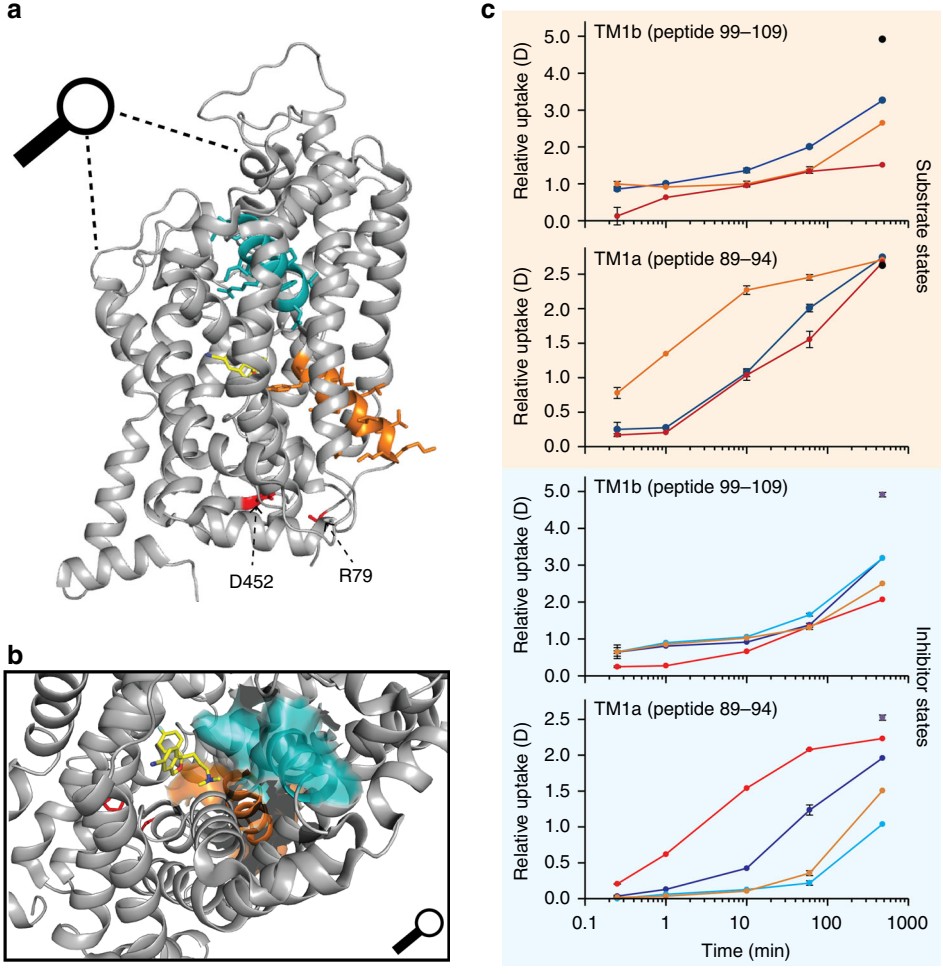

**Fig. 5** Differences in dynamics of TM1 in hSERT upon binding of ions, 5-HT and drugs (**a**). TM1a (orange), and TM1b (cyan) highlighted in the structure of hSERT (PDB: 5i75) along with Arg79 and Asp452 (residues shown red) which form the conserved salt bridge between the N-terminal domain and TM8, respectively. A magnifying glass gives an approximation of the angle for the zoom-in view of the central binding cavity. **b** Central binding cavity occupied by S-citalopram (carbons yellow, oxygen red, and nitrogen blue). In this zoom-in view, the surface of TM1a and TM1b are shown as orange and cyan, respectively. **c** Deuterium uptake plots for two peptides located in TM1a (peptide 89–94) or TM1b (peptide 99–109) in the presence of $Na^+$ (blue), $K^+$ (orange), and 5-HT (red) (brown box) or in the absence (blue, $Na^+$-reference state) and presence of the inhibitors cocaine (cyan), ibogaine (red), or S-citalopram (orange) (blue box). Error bars indicate standard deviations for time points measured in replicates ($n = 2$–4). Maximum-labelled controls are indicated with a black dot. Source data for the HDX data are provided as a Source data file

presence of $K^{+13}$. LeuT also showed significant differences in TM6 where no differences were seen in hSERT (Fig. 2), suggesting that $K^+$ binding also impacts the two transporters differently. Lack of sequence coverage in TM5 and TM7 of hSERT prevented comparison of these two regions with our previous analysis of LeuT.

An intriguing difference could be seen in the type of kinetics observed in the two transporters. EX1 kinetics were observed in numerous regions of LeuT (parts of TM1a, TM5, TM6, TM7, EL2, and EL4b), but were only visible in TM1b of $K^+$-bound hSERT. Based on the high structural similarities between hSERT and LeuT, it is unlikely that their mechanism of transport is dramatically different. The lack of widespread EX1 kinetics in hSERT (and thus inherently faster refolding dynamics) could correlate nicely with reported difference in the rate of substrate turnover ($k_{cat}$ (LeuT) = ~0.0003 s$^{-1}$[57]; $k_{cat}$ (hSERT) = 1–3 s$^{-1}$[58]).

We observed a similar HDX pattern in hSERT in the presence of $K^+$ and 5-HT (with $Na^+$), with destabilization in a segment preceding and including the beginning of TM1a (residues 70–88, Fig. 1a) and a stabilization of TM1b (Fig. 5), suggesting that both ligands induced similar conformational features. However, we

saw differences in TM1a and the extracellular loop regions (EL3 and EL4) between binding $K^+$ and 5-HT, revealing also differences in the conformations induced by these two ligands (Figs. 2, 3, and 5). These differences may be caused by $K^+$ promoting an inward-facing open conformation, while 5-HT may still be occupying the binding pocket (observed by reduced HDX for residues 89–95) in either a partial inward-facing open or closed conformation. Similar results have been obtained with unbiased MD simulations of SERT in complex with $Na^+$ and 5-HT[59]. They found that 5-HT binding leads to an isomerization of SERT to an inward-facing state, largely mediated by conformational changes in TM1a and 6b. Also in accordance with the present results, they propose that the two-halves of these helices are internally flexible with the intracellular part of the helices moving considerably more than the extracellular part.

hSERT in the presence of S-citalopram is reported to favor an outward-facing conformation[60], fitting with the crystal structure of the transporter[21]. Relative to the $Na^+$-bound state, we observe that S-citalopram binding only causes significant stabilization in a segment encompassing TM1 and a region preceding this helix (Figs. 4a, d and 5). According to the hSERT crystal structure[21]

and several MD simulations[24,60–62], S-citalopram is suggested to interact with both Tyr95 and Asp98 in TM1 but also with other SERT regions, including TM3 (Ile172 and Tyr176), TM6 (Phe335, Ser336, and Phe341), TM8 (Ser438 and Thr439), and TM10 (Thr497 and Val501). These regions (excluding Ser438, 439, and Val501, which were not possible to probe here) did not show a significant difference in HDX. Using the patch clamp technique on HEK cells expressing SERT, it has been suggested that the transporter already adopts an outward-facing open conformation merely by the presence of $Na^+$[20]. The increased structural stability observed throughout TM1 is then caused by S-citalopram interacting with Tyr95 and Asp98 as well as its bulky structure significantly hampering the structural fluctuations of the entire TM1 of hSERT. Similarly, the cocaine binding site could be formed by $Na^+$. This is supported by MD simulations showing that cocaine stabilized an outward-facing conformation, both in SERT[25] and DAT[49]. Thus, the observed stabilization by both inhibitors (Fig. 4) in TM1a, which includes Arg79 that forms a conserved salt bridge to Asp452 (Fig. 5), and the region immediately preceding it (residues 70–95) is likely to be important for trapping the transporter in an outward-facing conformation.

Nevertheless, a stabilization of TM1b was not observed in hSERT by the binding of cocaine. The difference in stabilization between the two inhibitors may be due to cocaine being less bulky than S-citalopram, allowing for more movement of TM1b. Experiments were in both cases performed at approximately 90% binding occupancy, thus the differences in affinity of the two ligands was taken into account as much as possible (see Methods section for further details). An alternative explanation is that S-citalopram, even at this relatively low concentration (340 nM), also associates to the S2 site causing a further stabilization of the molecule. However, the S2 site would be expected to only be occupied in around 6.5% of hSERT (IC$_{50}$ = 4.6 ± 0.4 μM[23]) present during HDX labeling. Also, we would expect S2 binding to cause a substantial stabilization of large stretches in multiple TM domains, which we do not observe here.

Out of the three inhibitors tested here, hSERT is only reported to favor an inward-facing conformation in the presence of ibogaine[55]. Like in the presence of $K^+$, ibogaine-induced structural destabilization was observed in TM1a and EL3 while TM1b and EL4 were stabilized. Though the conformational dynamics of hSERT in the presence of $K^+$ and ibogaine are similar, differences could be seen including the exchange kinetics in TM1b. Thus, with ibogaine, the TM1b region exchanges according to EX2 kinetics but in the presence of $K^+$ according to EX1 kinetics. Ibogaine binding has been reported to be non-competitive to 5-HT[54] and the lack of stabilization in the middle of TM1a (residues 89–95) seen with 5-HT but not for ibogaine, supports the hypothesis that ibogaine does not share a binding site with 5-HT.

All three ligands (i.e., $K^+$, 5-HT, and ibogaine) caused structural destabilization of the region preceding TM1a, TM1a itself and stabilization of TM1b, indicating this to be a signature of an inward-facing conformation in hSERT (Fig. 5). Other significant differences in the stability of hSERT varied between the three states. In particular, 5-HT still causes stabilization of TM1a (residues 89–95, (Fig. 5)). This could suggest that the transition from outward-to-inward has yet to be completed. MD simulations have supported this suggestion by elegantly showing that the solvent accessible surface area on the SERT intracellular side was higher in the noribogaine-bound transporter relative to 5-HT bound[25]. Another possibility is that our measurements are performed on hSERT in detergent and accordingly does not allow imposing a $Na^+$ gradient. This suggestion is supported by the stabilizing effect on EL4 by $K^+$ and ibogaine, but not 5-HT. In LeuT, EL4 plays an important role in closing access from the extracellular side[7]. If 5-HT binding only promotes but does not

complete the transition to the inward-facing conformation, the stabilization of EL4 might not be evident without the presence of a gradient.

Taken together, here we use HDX-MS to provide an unperturbed global view of the conformational dynamics of wild-type human SERT in solution and how such dynamics are modulated by binding of endogenous substrate (5-HT), co-transported ions ($Na^+/K^+$) and drug ligands (S-citalopram, cocaine, and ibogaine). Our results show that several loops and TM helices of the transporter undergo distinct changes in dynamics upon binding the substrate 5-HT and $K^+$ relative to the $Na^+$-bound state. Substrate-bound hSERT exhibits a clear difference in HDX relative to our previous observations on LeuT in the presence of leucine[15]. Thus, hSERT differs from LeuT in its conformational bias upon substrate binding, depicting limitations in using LeuT as a model protein for eukaryotic NSS proteins. By spatially resolving regions that change conformation in different functional and drug-bound states of hSERT, our results provide a framework for an improved understanding of the alternating access mechanism that governs the function of human SERT and could open up for novel drug strategies. Our findings should be impactful in refining our understanding of the conformational dynamics of hSERT and present/future structural models of distinct conformational states of hSERT (e.g., X-ray generated or based on computational models) will be supported by, and should account for, the experimental measurements on conformational dynamics of hSERT reported here. Considering that the transport mechanism and drug binding for SERT is dictated by its higher-order structure, we envisage that our results could be pertinent to other human transporters bearing the conserved NSS fold as these may operate via a similar mechanism of action.

## Methods

**Recombinant hSERT construct.** Full-length hSERT with a C-terminal thrombin site (LVPRGS) followed by a 12 histidine-tag was synthesized and cloned into the pEG BacMam expression vector[43] by Genscript Inc. (Piscataway, NJ). Further information on the recombinant hSERT construct can be found in the Supplementary Information.

**Expression and purification of human serotonin transporter.** Baculovirus for the expression of hSERT was produced using Sf9 cells (Expression Systems). The virus was used for the infection of mammalian Expi293F cells (Gibco)[43]. The Expi293F cells were incubated at 37 °C, 5% $CO_2$, 70% humidity, 130 rpm, in Expi293 Expression Medium (Gibco) until they reached density optimized for the expression of hSERT (here, 1.2–1.5 × 10$^6$ cells/ml culture). The cells were infected with 1% of P1 virus that was added along with 2 mM of valproic acid, and the cells incubated for two days at 37 °C, 5% $CO_2$, 70% humidity, 130 rpm, before harvesting. The cells were harvested at 6200$g$, frozen and stored at −80 °C. The cells were gently thawed and resuspended in cold lysis buffer (20 mM Tris (pH 8.0), 150 mM NaCl, 100 μM tri(2-carboxyethyl)phosphine (TCEP), 0.5 mM cholesteryl hemisuccinate (CHS), 30% glycerol, and protease inhibitors benzamidine and leupeptin, both in 10 μg/ml) and ruptured by sonication (Branson Sonifier 250 set at 50% duty cycle and output control 5). Non-lysed cells were removed by centrifugation (750$g$) for 15 min at 4 °C. Cell membranes were collected by centrifugation of the supernatant at 125,000$g$ for 2 h. The membranes were subsequently resuspended in solubilization buffer (20 mM Tris (pH 8.0), 150 mM NaCl, 500 μM TCEP, 4 mM CHS, 20 mM N-dodecyl-β-D-maltoside (DDM), 5 μg/ml benzamidine and 10 μg/ml leupeptin). The solubilized protein was purified by batch purification with single step nickel affinity purification and eluted with 20 mM Tris (pH 8.0), 300 mM NaCl, 500 μM TCEP, 300 mM imidazole, 0.5 mM CHS, 24 μM of lipid mixture (POPC:POPE:POPG; 1:1:1 ratio), 1 mM DDM, 5% glycerol, 5 μg/ml benzamidine, and 10 μg/ml leupeptin. The detergent and lipids added during SERT purification have been previously reported along with the structure of hSERT by Coleman et al.[21].

**Scintillation proximity assay for verification of activity.** The activity of hSERT was verified by saturation binding using scintillation proximity assay (SPA) performed largely as described previously[10]. hSERT at a final concentration of 0.4 μg/ml was added to a solution of YSi-Cu$^{2+}$ SPA beads (125 μg/ml, Perkin Elmer), and [$^3$H]imipramine in sample buffer (20 mM Tris (pH 8.0), 300 mM NaCl, 500 μM TCEP, 0.5 mM CHS, 24 μM of lipid mixture (POPC:POPE:POPG; 1:1:1 ratio), 1 mM DDM, 5% glycerol) was added in a concentration range (1–48 nM),

performed in triplicates. Unspecific binding (10 µM paroxetine) was determined in parallel. The binding experiment was incubated at room temperature for 30 min followed by overnight incubation at 4 °C to obtain equilibrium conditions. Binding activity was quantified on a MicroBeta scintillation counter (PerkinElmer). Following subtraction of the counts caused by non-specific binding, the data was plotted in GraphPad Prism version 6 and fitted to a one site saturation binding curve, allowing for the extraction of the $K_d$.

**Hydrogen/deuterium exchange**. Prior to HDX labeling, purified hSERT was dialyzed or diluted 1:4 (in the case of the HDX-MS experiments with inhibitors) into sample buffer (26 mM Tris (pH 8.0), 500 µM TCEP, 0.5 mM CHS, 24 µM of lipid mixture (POPC:POPE:POPG; 1:1:1 ratio), 1 mM DDM, 5% glycerol) containing either 300 mM NaCl or 300 mM KCl resulting in a final concentration of 2.16 µM. For HDX of hSERT in the presence of ligands, individual ligands were added as a part of the NaCl-containing sample buffer. 5-HT or inhibitors were added in a concentration that yielded a 90% binding occupancy of hSERT (0.31 µM) during the deuterium labeling reactions, based on published affinities (ligand concentrations: 10, 0.34, 2.4, and 25.5 µM, for 5-HT ($K_d$: ~1 µM)[44,63], S-citalopram ($K_d$: ~5 nM)[4,44,64], cocaine ($K_d$: ~220 nM)[4,44], and ibogaine ($K_d$: ~2.6 µM)[55,65], respectively). The HDX reactions were initiated by diluting the hSERT equilibration solutions 1:6.5 with deuterated buffer (26 mM Tris (pH 8.0), 500 µM TCEP, 0.5 mM CHS, 24 µM of lipid mixture (POPC:POPE:POPG; 1:1:1 ratio), 1 mM DDM with either 300 mM of NaCl or 300 mM of KCl, resulting in 85% deuterium content during HDX. HDX reactions were incubated for specific time periods (0.25, 1, 10 minutes, 60, and 480 min) at 25 °C before a sample was extracted from the reaction tube and quenched by 1:1 dilution into ice cold quench buffer (219 mM potassium phosphate (pH 2.5), 6 M urea, 1 mM DDM). Quenched samples were immediately frozen at −80 °C and stored until their intended measurements by liquid chromatography mass spectrometry (LC–MS).

We note that the HDX-MS ligand-binding experiments were performed using ligand concentrations which should ensure ~90% binding occupancy of hSERT, excluding effects caused by differences in affinity by any of the ligands. However, as the transporter was around 20–25% pure during HDX-MS (Supplementary Fig. 1), variability could occur in the binding occupancy of hSERT if the ligands displayed different unspecific binding to protein impurities present. Such a scenario would impact the exact magnitude of HDX differences observed, thus while our results reveal which regions of hSERT are perturbed by different ligands, a detailed quantitative comparison of HDX differences between different ligands should be approached with caution. Further, though purified from a non-neuronal cell line, some hSERT interacting proteins could remain in the sample. These possible interaction partners would however be present in all states (including the reference state) and any impact on hSERT should thus be the same in all HDX experiments. Additionally, in a control experiment where the HDX of the Na$^+$-bound reference state was performed following size exclusion chromatography (SEC), and thus under conditions with significantly fewer impurities, hSERT showed identical HDX, within the experimental error, to affinity-purified hSERT (Supplementary Fig. 2).

**Liquid chromatography and mass spectrometry**. Frozen quenched samples were quickly thawed in a table top centrifuge and injected into a refrigerated (0 °C) UPLC platform (NanoAcquity, Waters). The protein was passed through a self-pack column containing rhizopuspepsin immobilized on POROS 20AL resin at 20 °C. The generated peptides were trapped on a Vanguard column (C8, 130 Å, 1.7 µm, 2.1 mm × 5 mm; Waters) and desalted for 4 min at a flow rate of 150 µl/min solvent A (0.23% formic acid in MQ water, pH 2.5). Subsequently, the peptides were separated over an Acquity UPLC column (C8, 130 Å, 1.7 µm, 1 mm × 100 mm; Waters) with a linear gradient rising from 8 to 30% solvent B (0.23% formic acid in acetonitrile). Following the chromatographic separation, the peptides were analyzed using a hybrid ESI-Q-TOF mass spectrometer (Synapt G2-Si, Waters). The MS was set in positive ionization mode and the peptides were further separated by ion mobility for enhanced peak capacity. For peptide identification, nondeuterated samples were injected and subjected to identical chromatographic separation. The peptides were fragmented by collision-induced dissociation using argon as collision gas, either in data dependent (DDA) or data independent acquisition (DIA) mode.

**Maximum-labeled control samples**. Maximum-labeled controls were prepared by loading samples of hSERT on the same UPLC platform described above. However, following on-line rhizopuspepsin digestion, the peptides were trapped on a Vanguard trap column, desalted, eluted and manually collected. The mobile phase was evaporated using a SpeedVac (ThermoFisher). The pre-digested samples were resuspended in 85% deuterium buffer and incubated overnight before the samples were quenched as described for other HDX samples.

**Evaluation of stability**. Purified hSERT (50 µg in 300 µl) was injected onto an ÄKTA FPLC system with a Superose 6 Increase 10/300 GL. The separation was performed at 0.5 ml/min using sample buffer (20 mM Tris (pH 8.0), 300 mM NaCl, 500 µM TCEP, 0.5 mM CHS, 24 µM of lipid mixture (POPC:POPE:POPG; 1:1:1 ratio), 1 mM DDM, 5% glycerol) as a mobile phase. Fractions were collected from 4 ml until the end of the analysis. Fractions corresponding to hSERT were

upconcentrated from 2.5 ml to approximately 100 µl for HDX-MS analysis. For the evaluation of stability by HDX-MS, SEC-purified SERT was prepared with or without 1 h pre-incubation at 25 °C. The HDX labeling was performed over selected time intervals (0.25, 10, and 60 min) using HDX sample buffer containing 300 mM NaCl, and analyzed by LC-MS as described for other HDX-MS samples. Butterfly plots were prepared for comparison of individual states according to the following equation:

$$\text{Normalized HDX} = \frac{\text{HDX}_{\text{sample}}}{\text{HDX}_{\text{MX control}}}, \qquad (1)$$

where $\text{HDX}_{\text{sample}}$ is the deuterium uptake of a peptide at a respective measurement time point and $\text{HDX}_{\text{MX control}}$ is the experimentally determined maximum deuterium content of the peptide. Normalized deuterium uptake for two states was presented as either positive or negative values for each peptide. Negative values were calculated by multiplying the deuterium content of peptides with "−1" to obtain a mirror image for comparison.

**HDX-MS data analysis**. MS/MS data was processed with ProteinLynx Global Server (PLGS) version 3.0 (Waters), for peptide identification. Peptides identified by DDA had to have PLGS Ladder Score above 1.0, below 15 ppm mass error for the precursor ion, along with passing manual inspection of the fragment spectrum in PLGS. DynamX version 3.0 (Waters) was used to filter peptides identified by DIA with the peptides having to be fragmented at least 3 times, 0.1 fragmentations per amino acid, and below 10 ppm mass error for the precursor ion. Further, the peptides had to be identified in 2:3 of the acquired MS/MS files. Lastly, the presence of a signal for the precursor ion was verified in DynamX along with analysis of deuterium uptake of all positively identified peptides. Maximum labeled controls were used to calculate the back exchange (BE) for individual peptides by the following equation:

$$\text{BE} \, (\%) = \left( 1 - \frac{m_{\text{max}} - m_{0\%}}{N \, D_{\text{frac}}} \right) \times 100\%, \qquad (2)$$

where $m_{\text{max}}$ is the mass of a maximum labeled peptide, $m_{0\%}$ is the mass of a nondeuterated peptide, and $D_{\text{frac}}$ is the fraction of deuterium in the HDX labeling buffer and $N$ is the number of peptide amide hydrogens. Calculated BE values were used to support the identification of hSERT peptides and to validate HDX-MS system performance. The plotted deuterium uptake profiles were not corrected for BE.

Preliminary HDX-MS screening experiments ($n = 1$) were performed on hSERT states with all time points (0.25, 1, 10, 60, 480 min). Based on the preliminary experiments, the final HDX-MS data set was obtained from a series of comprehensive HDX-MS experiments performed on fresh batches of hSERT ($n = 3$ biological replicates) with technical replicate measurements obtained for selected time points ($n = 2$–4 technical replicates).

To determine whether a difference in HDX between two states was significant or not, the standard deviation of time points performed at least in triplicate was used to calculate a confidence interval. The standard deviation of individual peptides ($n = 3$) was averaged across the states being compared. The average standard deviation was then used to calculate the accumulated standard deviation and the 95% confidence interval (CI)

$$\sigma_{\text{ave}} = \sqrt{n \times \bar{\sigma}^2}, \qquad (3)$$

$$\text{CI} = \bar{x} \pm t \times \frac{\bar{\sigma}_{\text{ave}}}{\sqrt{n}}, \qquad (4)$$

where $\sigma_{\text{ave}}$ is the accumulated average standard deviation, $\bar{x}$ is the assumed zero-centered average difference in HDX between two states (i.e., $\bar{x} = 0$), t is the table value for the two-tail 95% CI with two degrees of freedom ($t_{95\%, \, n=3} = 4.303$), $\bar{\sigma}_{\text{ave}}$ is the averaged standard deviation for all measured peptides, and $n$ is the number of replicates ($n = 3$). In the comparative HDX analysis of the different functional states, a peptide was only considered to have a significant difference in HDX if two consecutive time points showed a significant difference in deuterium incorporation above the average CI or based on a Student's $t$-test ($p < 0.01$). As an exemption to this rule, a difference was also considered significant if the last time point (480 min) displayed a difference in HDX at least two times larger than the CI, as this indicates a significantly expanding difference not fully sampled by the HDX time course.

To allow access to the HDX data of this study, the HDX Data Tables (Supplementary Tables 1–3) and the HDX Data Summary Table (Supplementary Table 4) are included in the Supporting Information according to the community-based recommendations[66].

**Reporting summary**. Further information on experimental design is available in the Nature Research Reporting Summary linked to this article.

## Data availability

Data supporting the findings of this manuscript are available from the corresponding authors upon reasonable request. A reporting summary for this Article is available as a Supplementary Information file. The source data underlying Figs. 1, 2, 3, 4, 5c, and Supplementary Figs. 1, 2, 5, 6, 7, 8, are provided as a Source Data file. Mass spectrometry

data files including processed DynamX files and an overview of the HDX-MS data (Supporting Tables 1–4) have been uploaded to the PRoteomics IDEntfication (PRIDE) Database.

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

## Acknowledgements

This work was made possible through support from the Danish Council for Independent Research (0602-02740B to K.D.R., 0602-02100B and 4183-00581 to C.J.L.), the UCPH bioSYNergy center of excellence and the Novo Nordic Foundation (NNF17OC0028582 to C.J.L.), and the Lundbeck Foundation (SGFR). The pEG BacMam vector was kindly provided by Eric Gouaux, Ph.D., at the Vollum Institute. Thanks to Giuseppe Cazzamali, Novo Nordisk Foundation Center for Protein Research, for kindly providing access to facilities allowing SERT expression in HEK293expi cells. Rhizopuspepsin used for HDX-MS as well as other proteases screened in this work were kindly provided by Petr Man at BioCev, Czech Academy of Sciences.

## Author contributions

K.D.R. and C.J.L. conceived and supervised the work. I.R.M. produced recombinant S.E.R.T. with the help of M.S. and A.K.N. I.R.M. performed the H.D.X.-M.S. experiments. I.R.M. and M.S. performed the S.P.A. experiments. I.R.M. and K.D.R. analyzed the H.D.X.-M.S. data. C.J.L., K.D.R., S.G.F.R., and U.G. contributed with equipment, reagents, and knowhow. I.R.M., C.J.L., and K.D.R. prepared the paper and all authors read and commented it.

## Additional information

**Competing interests:** The authors declare no competing interests.

