## [Peer Review File · Nature Communications]

Reviewers' Comments:

Reviewer #1:

Remarks to the Author:

Möller et al. present an interesting manuscript reporting on the application of hydrogen-deuterium exchange mass spectrometry (HDX-MS) on the serotonin transporter (SERT), which is quite an impressive undertaking. Although many functional states represented by different structures are known for the SLC6 family, it is also evident that for example SERT displays unique properties such as K⁺ counter-transport and differential effects of various inhibitors, such as S-citalopram, cocaine, and ibogaine that also involve an extracellular site outside of the substrate site (serotonin/5-HT). The study analyses intramolecular dynamics associated with different states and state transitions. There are some technical considerations and questions as listed below, but overall the study bears merit, addresses important questions, and provides some interesting answers.

1. The protein is not very pure (suppl. fig. 1). Did the authors try a cleavage of the C-terminal tag (seemingly not) followed by another round of reverse Ni²⁺ purification, i.e. capturing all impurities again, but not the released SERT?
2. The impurities among other concerns may affect the accurate titration at 90% occupancy and therefore state-to-state comparisons. Even though e.g. S-citalopram and 5-HT binding is supposedly specific and of high and medium-high affinity, respectively (as determined by the SPA assay) it may still be flawed by unspecific binding. It depends on which background control samples were used for subtraction of unspecific binding. SERT-free membranes would be the right control, but this is not stated in the methods
3. The use of a SERT sample with a seemingly uncleaved C-terminal tag also questions the results and discussions on the C-terminus and the effect of imidazole and K⁺ on C-terminal dynamics – this may be heavily influenced by the long extension to the C-terminus with a Thrombin-site followed by the (His)₁₂ tag, which is likely to be very sensitive to chelators or trace amounts of e.g. Zn²⁺ and Ni²⁺ (e.g. lines 301-9 in the discussion)
4. The coverage is maximal with rhizopuspepsin, which is therefore used, but did other pepsins produce complementing, unique fragments that combined with the rhizopuspepsin fragmentation could have provided a better coverage and comprehensive analysis of key elements?
5. The introduction returns at many occasions to LeuT studies. Although relevant to mention structural studies of LeuT and other SLC6 transporters and also earlier HDX-MS studies of LeuT, there is no reference provided for the latter (lines 70-72), which therefore does not read well, but rather appears like an introduction to work performed in this study. Next when K⁺ binding is introduced it should be focused on SERT and not start out with a reference to own work on LeuT, which is less relevant (if at all) on this particular matter (lines 75-78)
6. Dynamics have been extensively studied with MD simulations in many groups, and there are also many studies probing SERT dynamics and conformations with reagents. The sentence “current knowledge about the conformational dynamics of SERT is limited and is based on inferences from biochemical and mutagenesis studies²³⁻²⁵” (lines 88-89) therefore seems inappropriate.
7. Discussion: opening sentence on available structures “major truncations, mutations or deletions have been required for the crystallization of these proteins” (lines 266-67) needs to go out or be more specific - it does not apply to all structures, only some, and it is therefore an unfair justification for the study.

Reviewer #2:

Remarks to the Author:

Rand and colleagues report on the conformational dynamics of the human serotonin transporter, SERT, upon ion, substrate and inhibitor binding. They carry out extensive hydrogen-deuterium exchange mass spectrometry measurements under different conditions and the results highlight interesting commonalities and differences between structural responses upon binding. This is an

interesting study based on a rigorous experimental design and application. Moreover, the study of eukaryotic transporters is highly underrepresented in the literature as they pose significant challenges experimentally. There is however a lack of context from the relevant literature that undermines the overall message. Moreover, a few controls should be carried out to ensure that the protein is folded and not aggregated during the HDX-MS experiments. Overall, the study will be of general interest and should be published after the suggested controls are carried out and the structure of the article is simplified. See specific suggestions below.

Major comments:

1. In general, the paper suffers from a lack of context from the existing literature. It appears that the experiments and findings are interesting and potentially useful for a wide range of scientists. However, as it stands, it is hard to place the experimental findings within the existing structural and functional context of SERT protein. Moreover, in parts it feels like that the results from HDX-MS lack biological interpretation. While the authors present their data in a very detailed and clear fashion, the reader is left wondering on how these result may be useful in further understanding the structural and functional mechanism of the system in question. Nonetheless it should be noted that this does not diminish the importance of reporting the finding of SERT dynamics.

2. Along these lines, it appears from the HDX data that TM1 plays a very important role and seems to be the main marker of conformational changes. This is an interesting feature of the paper and may be associated with the mechanism of binding and conformational transition between different states. As such, could the authors provide some background to this striking feature? I believe a few biochemical studies on MD simulations have been performed which could provide interesting structural and functional context (eg: doi: 10.1124/mol.115.099911, <https://doi.org/10.1371/journal.pone.0063635>). Furthermore, wouldn't it make sense to add a section either in the results or the discussion section that focuses on that helix and provide a comparative analysis for all conditions studied here? Looking at the data, it's clear that K⁺, 5-HT and ibogaine binding, all induce similar structural shifts, while cocaine and citalopram present an opposite pattern.

3. The gel in the supplementary figure 1 shows degradation/impurities and this raises question about the stability of the transporter. Is the protein still folded after 60 min of deuteration at room temperature? The authors should perform Circular Dichroism or FTIR experiments on the purified transporter to ensure that the secondary structural elements are still present. Have the authors carried out Size Exclusion chromatography to ensure the purified protein was homogenous and aggregation was limited? In line with this comment, the fact that the authors see high levels of HDX after 30 sec in TM regions might suggest that parts of the proteins are unfolded. I understand that the functional assay was performed to demonstrate that the protein was in its native functional state but the lack of purity of the sample raises doubt regarding the results.

4. The presence of a mix of lipids throughout the whole experiment is intriguing. Could the authors explain why they add lipids? Is it a common practice for hSERT purification? Does it contribute to the stability of the protein? The presence of lipids is not trivial because it has been shown to modulate the conformational dynamics of other transporters and receptors, even in the absence of a bilayer (doi: 10.1038/nchembio.1960, doi: 10.1038/nsmb.3262). The authors might want to justify the use of lipids and discuss this further.

5. A recent study by Anderluh A, et al, Nat Commun. 8, 14089, pointed to the formation of SERT oligomers. The study suggested the arrangement allows for formation of both dimers and other higher order oligomers. This oligomeric formation likely to be mediated by lipids. However, molecular level details of the interaction between the lipid and SERT have yet to be elucidated. Do the authors have any evidences of oligomeric formation? Or are they certain of the monomeric arrangement for SERT? If SERT oligomerises, the presence of more than one monomers may introduce asymmetry in the reported HDX data. The authors may want to discuss this issue in their

paper.

Other concerns:

- What is the meaning of "ligand-attenuated changes"?

Introduction:

- There is a lack of references overall, e.g.: crystal structure of LeuT, pharmacological uses of SSRIs, etc. Please acknowledge previous work.

- Line 77: "a role of K⁺ has been shown for SERT function". Be specific

- Line 80: "functional" conservation should be replaced by "mechanistic" – NSS transport different substrates so they don't have the same function – but they might do it in a similar way.

Results

- I'm intrigued by the use of rhizopuspepsin. Could the author mention how they obtained that type of pepsin? What benefits does it offer?

- The authors use a 95 confidence interval that is not very stringent. The authors might consider using a 98 or even 99 CI and see whether they have more consistency between the region reporting conformational changes in the different experiments. For example, in figure 2, the authors include on the figure the peptide 483-499, but looking at the uptake plot, the difference seems quite small and it might be insignificant. The authors actually seem to share this opinion since they don't mention the peptide in the main text. In the same paragraph, the authors state that the overall uptake pattern is consistent with an intracellular opening coupled to an extracellular closing upon K⁺ binding. This doesn't seem a statement representing the data on fig.1 as we see uptake on both sides, but using a more stringent confidence interval might prove them right.

- Line 175. I'm not sure I follow the line of thoughts of the authors when they claim that the observed EX1 kinetics on TM1b correlates with a closing of the extracellular side. Could they be more explicit and explain this further? Overall there is more uptake in the presence of K⁺.

- Line 178: Why do the authors control the role of imidazole on HDX if they exchange buffer overnight by dialysis? There shouldn't be any imidazole around anymore.

- The authors observe important HDX in regions that are supposedly α -helices and are expected to undergo slow HDX. Can the authors ensure that the protein is folded?

- It would be helpful to annotate the topological maps on the figures with helices number etc

- Why is peptide 217-235 coloured in fig.3? it doesn't seem that anything is going on there when looking at the uptake plots. Again, there is no mention of this peptide in the main text anyway.

- Lines 198-199: It sounds like an over-claim that changes induced by 5-HT probably indicate the inward-facing state. It appears to me that mixed signals are obtained from HDX-MS that are difficult to attribute to a specific state. The authors should modify this statement accordingly.

- Line 200: I'm not sure it makes sense to compare the K⁺ state with the Na⁺/5-HT state.

Discussion:

- Line 273: Replace "native" by "in-solution". The absence of a bilayer, a gradient and whatever other components that are part of an intact cell preclude the use of that word.

- Line 284: Upon K⁺ binding

- Line 301: I question the significance of the difference observed for the C-terminal peptides. What happens if you use a higher confidence interval?

- Line 315: "these regions did not show a significant difference in HDX". Obvious question but did these regions display any HDX at all? If they are too protected, by the micelle for example, I'm not surprised.

- Line 343: I agree that the HDX behaviour of TM1b seems to be a hallmark of the conformational a transition and I would suggest presenting it in a cartoon.

-

Figures

The figures are overall of high quality and clear to follow. What is somewhat counterintuitive is the presentation in Figures 2A, 3A and 4A-C. I believe that the authors plot the Delta HDX of (Na⁺ - ion/substrate/inhibitor state). This is opposite of what it is plotted in panes 2B, 3B and 4D-F. It will therefore be more consistent if they will plot Delta HDX of (ion/substrate/inhibitor state - Na⁺ state).

Supplementary

- I am surprised by the statement that the protein was deemed pure enough. The gel doesn't show a pure protein. This raises questions about the results of the binding assay and the overall stability of the protein.

- Thanks for providing the uptake plots, that's very useful.

Reviewer #3:

Remarks to the Author:

The authors examine the dynamics and substrate/drug-binding of the human serotonin transporter (hSERT) using hydrogen/deuterium exchange mass spectrometry (HDX-MS). hSERT is a highly impactful subject of investigation due to its role as target of therapeutics for the treatment of psychiatric disorders. Investigating the dynamics and conformational changes of a membrane-embedded protein remains extremely challenging. The authors have surmounted these challenges to present a detailed, spatially-resolved portrait of SERT dynamics when bound to substrates Na⁺, K⁺, and serotonin. Furthermore, the authors probe the structural perturbations induced by drugs of clinical and societal importance (i.e., the SSRI S-citalopram, cocaine, and ibogaine). Ground-breaking crystallography-derived structural models of SERT have been recently reported. However, the extensive stabilization/solubilization strategies required for x-ray crystallography can obscure functionally critical dynamics and conformational landscapes. The authors directly address this missing dimension in this investigation by probing solution dynamics on full-length, native hSERT. Despite the challenges of working with a transmembrane protein, this work achieves good coverage with high technical rigor. The authors integrate their dynamics observations with pre-existing biochemical literature. Importantly, many new insights are offered. For example, the authors correlate functional distinctions with dynamics differences between hSERT and LeuT, another member of this transporter family (lines 199-207). In addition, there are examples of previously unknown allosteric changes associated with drug binding (e.g., lines 218-222). Intriguingly, these studies reveal opposing effects on local dynamics induced by different drugs (e.g., ibogaine vs. cocaine and S-citalopram).

Statistical analysis is exhaustively reported and appropriate, and the experiments were performed with both technical and biological replicates. Methods for peptide identification by MS are stringent and well-reported. Regarding reproducibility, the authors have included excellent technical notes about the optimization of the HDX methods in the Supporting Info. These notes include rarely-reported details about factors that failed to improve data quality. Given the extremely wide interest in hSERT pharmacology, the technical groundwork established and reported here will ensure that others in the field will also be able to make use of these strategies when investigating

the many other current and future drugs targeting hSERT.

I am enthusiastic about this work. I just have a few minor comments to improve clarity.

Minor Points:

The authors present an SDS-PAGE gel to illustrate the hSERT sample used for subsequent analysis. The protein sample is highly contaminated with other species. On the one hand, this is an excellent demonstration of the robustness and sensitivity of HDX-MS. However, I recommend a few cautionary notes regarding sample homogeneity:

- a. There are several mentions of “purified” protein, yet the gel indicates that “enrichment/enriched” are more accurate terms.
- b. The authors note regulatory interactions with SERT (line 306). Given that the SERT protein was expressed and purified from a non-neuronal, recombinant source (i.e., an embryonic kidney cell line), I am largely satisfied that the contaminating proteins are unlikely to interfere or interact with SERT. However, it would be worth pointing out this caveat explicitly in the text somewhere.
- c. Notably, contamination with SERT proteolysis products (from endogenous proteases liberated during cell lysis) is a potential risk in non-homogenous protein samples. The HDX data does not seem consistent with significant contamination with proteolyzed SERT. Nevertheless, the authors could clarify the sample quality by Western blotting with a polyclonal antibody for hSERT or an anti-His tag antibody.

The HDX was performed on DDM-detergent-solubilized SERT. This approach appears to be standard in the field, and the authors include binding data demonstrating that the protein retains function with DDM. However, micellar detergent “belts” may influence structure differently than the native planar bilayer. Generally, this variable is isolated, as the authors use the Na⁺-bound, DDM-solubilized sample as a baseline control. There is one location in the Results where this potential difference between solubilizing micellar lipids vs. planar bilayer should be addressed. The paragraph describing intrinsic dynamics (lines 140-145) notes unusually fast exchange rates in TM helices evident in the crystal structure. The authors should include the non-native solubilization conditions as a possible alternative explanation for these exchange rates.

Figs 1-4 share a view of the SERT structural model, upon which the exchange perturbations are mapped. It may clarify the topology of the protein in the membrane to annotate Fig. 1 with labels for the “extracellular” and “intracellular” sides. Consider also an estimation of the bilayer position with dotted lines denoting the lipid headgroup layers.

On lines 193-194, the authors point out a functionally important salt bridge informed on by the HDX results. A small figure (an inset or SI figure, perhaps) illustrating the side chain interaction with the HDX data mapped would be helpful.

There is a brief interlude (lines 178-184) wherein the authors compare whether the presence of imidazole influences exchange dynamics in the presence of Na⁺ vs. K⁺. I’m not entirely clear on why this is included in the manuscript, and a brief note of justification beyond “as a control” would be helpful. Perhaps it could be relegated to the Supporting Info? Minimally, the concentration of imidazole used should be included in the figure (SI Fig. 4).

Supplementary Fig. 1 includes a BSA standard curve in the SDS-PAGE gel. Given the use of gel staining for protein quantification, the staining technique used should be stated in the figure legend.

Supplementary Fig. 4: I am unclear about the meaning of “summed deuterium uptake” to describe the gray bars. Can the authors clarify?

A few typos: line 265, “outmost”?; line 130 “...and at least other 12% form...”

REVIEWER 1

Comment #1:

The protein is not very pure (suppl. fig. 1). Did the authors try a cleavage of the C-terminal tag (seemingly not) followed by another round of reverse Ni²⁺ purification, i.e. capturing all impurities again, but not the released SERT?

Our response: We thank the reviewer for the opportunity to clarify. Mammalian NSS proteins, like SERT, are notoriously difficult to express and purify and the suggested procedure was not attempted. The reasoning was that the transporter exhibited the expected binding profile ($K_D = 7.5 \pm 0.7$ nM; Supplementary Figure 1c) from the literature (Talvenheimo et al. (1983; PMID: 6853478); DOI: 10.1021/bi048022b) without removing the C-terminal tag and the extra steps would lead to unwanted loss of protein and thus dilution of the protein sample. This would have proven problematic to the subsequent HDX-MS experiments as the protein was already at quite low concentration following the first Ni²⁺-NTA affinity purification.

We have performed additional control experiments to assess the nature of impurities in the SERT sample and can confirm that these are not SERT degradation products (please see our response below to Comment #2 of the same reviewer for details).

Comment #2:

The impurities among other concerns may affect the accurate titration at 90% occupancy and therefore state-to-state comparisons. Even though e.g. S-citalopram and 5-HT binding is supposedly specific and of high and medium-high affinity, respectively (as determined by the SPA assay) it may still be flawed by unspecific binding. It depends on which background control samples were used for subtraction of unspecific binding. SERT-free membranes would be the right control, but this is not stated in the methods.

Our response: We welcome the change to clarify. Some unspecific binding is usually present in radioligand binding assays and a control for unspecific binding was performed using high concentrations of another high affinity ligand, in this case paroxetine ([Paroxetine] = 10 μ M; K_D = 0.13 nM (DOI: 10.1016/S0014-2999(97)01393-9)). With the ligands sharing the same binding site and with a high concentration of the higher affinity, non-radiolabeled paroxetine, we are convinced that any residual signal can be used to accurately quantify the unspecific binding of [³H]-imipramine. We believe this to be standard method and superior for background subtraction relative to performing a purification procedure of SERT-free membranes. The over-expression of SERT causes a considerable change in cell composition and hence the membranes will not be comparable.

It is stated in the Methods section on p. 20:

“Unspecific binding (10 μ M paroxetine) was determined in parallel... Binding activity was quantified on a MicroBeta scintillation counter (PerkinElmer). Following subtraction of the counts caused by non-specific binding...”

Furthermore, we have now directly investigated the occurrence of SERT degradation products in the SERT sample used for HDX-MS and western blotting analysis reveal these to be negligible.

This additional data has now been included in the Supporting Information and are referred to in the main text on p. 7:

“SDS-PAGE analysis of the purification indicated around 20-25% purity and the absence of SERT proteolysis products in the hSERT sample used for HDX-MS was confirmed by western blotting analysis (Supplementary Fig. 1).”

Comment #3:

The use of a SERT sample with a seemingly uncleaved C-terminal tag also questions the results and discussions on the C-terminus and the effect of imidazole and K⁺ on C-terminal dynamics – this may be heavily influenced by the long extension to the C-terminus with a Thrombin-site followed by the (His)₁₂ tag, which is likely to be very sensitive to chelators or trace amounts of e.g. Zn²⁺ and Ni²⁺ (e.g. lines 301-9 in the discussion).

Our response: We thank reviewer for the opportunity to elaborate on this concern also discussed in our reply to comment #1. We agree that the magnitude of the change in dynamics in the C-terminal segment upon ion or ligand binding (e.g. K⁺ and 5-HT binding) could be impacted by the His-tag. However, the fact that the dynamics is substrate-modulated cannot be explained simply by the presence of the His-tag and thus we are convinced that the dynamics of parts of the C-terminal segment of SERT are also substrate-modulated (to some extent) in SERT without the His-tag. We agree with the reviewer that there may be trace amounts of Zn²⁺ and Ni²⁺ in our SERT preparation and in the buffer, but that also means that it is present with and without ligand (as these were also prepared in the same buffer as the protein).

Furthermore, we can add that we have, in response to concern #6 made by Reviewer 2, now adopted a more conservative threshold for assigning significant differences in HDX. As a result, the effects observed in the part of the C-terminal of SERT that are most proximal to the His-tag (mentioned here by Reviewer #1) are no longer significant. Now, the only difference in HDX in the C-terminal region of SERT is confined to residues within TM12 (residues 595-598 within the peptide spanning residues 595-607). The results and discussion have now been modified accordingly.

As an example in p.8, lines 157-159 (lines 169-171 in revised version), the overlined deletions were made (“and the C-terminal helix” was removed and “residues 595-607” was changed to “residues 595-608”):

“Increased HDX was observed on the intracellular face in residues located in the N-terminus (residues 70-83), TM1a (residues 84-95), EL3 (residues 296-305 and 312-320), along with the intracellular parts of TM12b and in the C-terminal helix (residues 595-607-598).”

Comment #4:

The coverage is maximal with rhizopuspepsin, which is therefore used, but did other pepsins produce complementing, unique fragments that combined with the rhizopuspepsin fragmentation could have provided a better coverage and comprehensive analysis of key elements?

Our response: To some extent. During the screening of HDX-MS-compatible proteases, we observed that the sequence coverage could be increased by approx. 10% by combining results from use of complementary proteases. However, following the screening phase, the sample preparation was further optimized with regards to digestion with rhizopuspepsin to further the sequence coverage obtained with only this protease (e.g. addition of detergent in the quench buffer). Thus we do not expect that the same increase in sequence coverage could be realized using data from two parallel HDX-MS experiments with the same sample preparation but using rhizopuspepsin and another protease. Furthermore, even if this was the case, we do not believe this minor increase in coverage would justify the cost in terms of increased sample consumption of the small amount of SERT available for the study.

Comment #5:

The introduction returns at many occasions to LeuT studies. Although relevant to mention structural studies of LeuT and other SLC6 transporters and also earlier HDX-MS studies of LeuT, there is no reference provided for the latter (lines 70-72), which therefore does not read well, but rather appears like an introduction to work performed in this study. Next when K⁺ binding is introduced it should be focused on SERT and not start out with a reference to own work on LeuT, which is less relevant (if at all) on this particular matter (lines 75-78).

Our response: We thank the reviewer for this comment. In lines 70-72, we link our previous work on LeuT with this current work on SERT solely to introduce the reader to the relevance of using HDX-MS to study a structurally related model protein. We agree however that this does not read well.

To address the valid point by the reviewer, we have now inserted the requested reference (Merkle et al., 2018; DOI: 10.1126/sciadv.aar6179) on lines 70-72 (lines 67-69 in revised version) and rephrased the text on lines 75-78 (lines 72-77 in revised version) as follows:

“A role of K⁺ has also been shown for SERT function, that is K⁺ binds to the inward-facing transporter and is transported out of the cell, rendering the transport charge-neutral²² and speeding up the return-step of the transport cycle²³. Interestingly, recent findings suggest a role of K⁺ also in LeuT. For this transporter, it was found that K⁺ inhibits Na⁺ and substrate binding, and biases the transporter towards an inward facing conformation, thereby stabilizing the extracellular domains and destabilizing the intracellular face^{16,18}.”

Comment #6:

Dynamics have been extensively studied with MD simulations in many groups, and there are also many studies probing SERT dynamics and conformations with reagents. The sentence “current knowledge about the conformational dynamics of SERT is limited and is based on inferences from biochemical and mutagenesis studies²³⁻²⁵” (lines 88-89) therefore seems inappropriate.

Our response: The reviewer is correct that the molecular dynamics of LeuT have been studied by MD simulations in several groups. However we do believe that our knowledge from experiments of the conformational dynamics of SERT is quite incomplete and largely based on biochemical probes and mutagenesis studies that require covalent modification of the protein which can introduce non-native artefacts.

To accommodate the request, we have now rephrased the relevant section on page 5 as follows:

“However, in spite of the apparent structural similarity between LeuT and SERT, and several MD simulations of SERT²⁵⁻²⁷, our knowledge from experimental data on the conformational dynamics of SERT is incomplete and based on indirect inferences from biochemical and electrophysiological studies²⁸⁻³² often using SERT mutants.”

Comment #7:

Discussion: opening sentence on available structures “major truncations, mutations or deletions have been required for the crystallization of these proteins” (lines 266-67) needs to go out or be more specific - it does not apply to all structures, only some, and it is therefore an unfair justification for the study.

Our response: We thank the reviewer for this comment. The reviewer is correct that the sentence was not correct with regards to many other structures of family members such as LeuT. We have rephrased the sentence on page 13 to focus on the eukaryotic NSS proteins (dDAT and hSERT), for which the structures have been modified:

“However, truncations, mutations or deletions have been required for the crystallization of the eukaryotic NSS proteins, sometimes even rendering them incapable of transport.”

REVIEWER 2

Comment #1:

In general, the paper suffers from a lack of context from the existing literature. It appears that the experiments and findings are interesting and potentially useful for a wide range of scientists. However, as it stands, it is hard to place the experimental findings within the existing structural and functional context of SERT protein. Moreover, in parts it feels like that the results from HDX-MS lack biological interpretation. While the authors present their data in a very detailed and clear fashion, the reader is left wondering on how these result may be useful in further understanding the structural and functional mechanism of the system in question. Nonetheless it should be noted that this does not diminish the importance of reporting the finding of SERT dynamics.

Our response: Firstly, our HDX-MS results provide the first comprehensive view of the solution-phase dynamics of SERT and how these dynamics change when the transporter interacts with biologically relevant ions and substrate. Furthermore, we reveal for the first time the impact on the dynamics of SERT upon binding key therapeutic, illicit and psychedelic drugs (S-citalopram, cocaine and ibogaine, respectively). We are thus convinced that the work will be highly impactful in framing and refining our understanding of the dynamics of SERT and we believe that all static structural models of SERT (x-ray generated or based on computational models) will be enhanced by, and need to account for, the spatially-resolved experimental measurements reported here.

We specifically discuss this in detail throughout the discussion and frame our findings wrt. existing structural and functional knowledge on SERT (see for instance, lines 278-287, 298-312, 314-322, 346-358. However, we welcome the opportunity to further elaborate on how our HDX-MS results further understanding of the structure/function relationship of SERT and we have now added to and revised (highlighted in yellow) the following part of the discussion to accommodate the request of the reviewer:

Line 282 (294 in revised version):

“Compelling data suggest that K⁺ is counter-transported relative to the inward transport of Na⁺ and 5-HT₅₇, which could support a bias towards a more inward-facing structure by K⁺. The impact of K⁺ on hSERT revealed in this study clearly support this hypothesis and maps which regions are involved in this transition (peptides spanning residues 70-89, 89-95, 99-109, 296-305, 312-329, 397-406, and 400-406). LeuT was also shown to favor an inward-facing conformation in the presence of K⁺ 16. LeuT also showed significant differences in TM6 where no differences were seen in hSERT (Fig. 2), suggesting that K⁺ binding also impacts the two transporters differently.”

Line 298 (313-320 in revised version):

“We observed a similar HDX pattern in hSERT in the presence of K⁺ and 5-HT (with Na⁺), with destabilization in a segment preceding and including the beginning of TM1a (residues 70-88, Fig 1a) and a stabilization of TM1b (Fig. 5), suggesting that both ligands induced similar conformational features. However, we saw differences in TM1a and the extracellular loop regions (EL3 and EL4) between binding K⁺ and 5-HT revealing also differences in the conformations induced by these two ligands (Fig. 2, 3 and 5). These differences may be caused by K⁺ promoting an inward-facing open conformation while 5-HT may still be occupying the binding pocket (observed by reduced HDX for residues 89-95) in either a partial inward-facing open or closed conformation.”

Line 373 (391 in revised version):

“Our findings should be impactful in refining our understanding of the conformational dynamics of hSERT and present/future structural models of distinct conformational states of hSERT (e.g. X-ray generated or based on computational models) will be supported by, and should account for, the experimental measurements on conformational dynamics of hSERT reported here.”

Comment #2:

Along these lines, it appears from the HDX data that TM1 plays a very important role and seems to be the main marker of conformational changes. This is an interesting feature of the paper and may be associated with the mechanism of binding and conformational transition between different states. As such, could the authors provide some background to this striking feature? I believe a few biochemical studies on MD simulations have been performed which could provide interesting structural and functional context (eg: doi: 10.1124/mol.115.099911, <https://doi.org/10.1371/journal.pone.0063635>). Furthermore, wouldn't it make sense to add a section either in the results or the discussion section that focuses on that helix and provide a comparative analysis for all conditions studied here? Looking at the data, it's clear that K⁺, 5-HT and ibogaine binding, all induce similar structural shifts, while cocaine and citalopram present an opposite pattern.

Our response: We thank the reviewer for this useful suggestion. In addition to changes made to accommodate the related comment #1 of the reviewer, we have now added the following text to the discussion and inserted a new figure (Figure 5) that performs a specific comparison of the conformational effects on TM1 induced by ions, substrate and drug ligands. Furthermore, we reference 1 of the 2 papers suggested by the reviewer (reference #26). Further description of the differences observed in TM1a and TM1b has now been added in the discussion (p.16. line 353-373 in revised version):

“All three ligand-states which are reported to promote an inward-facing conformation in hSERT (i.e. K⁺, 5-HT, and ibogaine) caused structural destabilization of the region preceding TM1a, TM1a itself and stabilization of TM1b, indicating this to be a signature of an inward-facing conformation in hSERT (Fig. 5). Changes to the stability of these regions is in agreement with what has been observed for LeuT^{18,19}. Oppositely, the entire TM1 was stabilized in the presence of S-citalopram but only TM1a was stabilized in the presence of cocaine. As both inhibitors are known to promote an outward-facing conformation in hSERT^{26,60}, the observed stabilization in TM1a (Fig. 5), which includes Arg79 that forms a conserved salt-bridge to Asp452, is likely to be important for trapping the transporter in an outward-facing conformation.”

“Other significant differences in the stability of hSERT varied between the three inward-facing conformational states. Thus, while hSERT is thought to favor an inward-facing open conformation in the presence of K⁺ and ibogaine⁵⁵, 5-HT still causes stabilization of TM1a (residues 89-95, (Fig. 5)), suggesting that the transition from outward-to-inward has yet to be completed. This fits with reported MD simulations, where hSERT did not reach a fully inward-facing conformation when simulated in the presence of 5-HT while the noribogaine-bound transporter reached a fully inward-facing conformation during the simulation²⁶. Another possibility is that our measurements are performed on hSERT in detergent and accordingly does not allow imposing a Na⁺ gradient. This suggestion is supported by the stabilizing effect on EL4 by K⁺ and ibogaine, but not 5-HT. In LeuT, EL4 plays an important role in closing access from the extracellular side¹⁰. If 5-HT binding only promotes but does not complete the transition to the inward-facing conformation, the stabilization of EL4 might not be evident without the presence of a gradient.”

Comment #3:

The gel in the supplementary figure 1 shows degradation/impurities and this raises question about the stability of the transporter. Is the protein still folded after 60 min of deuteration at room temperature? The authors should perform Circular Dichroism or FTIR experiments on the purified transporter to ensure that the secondary structural elements are still present. Have the authors carried out Size Exclusion chromatography to ensure the purified protein was homogenous and aggregation was limited? In line with this comment, the fact that the authors see high levels of HDX after 30 sec in TM regions might suggest that parts of the proteins are unfolded. I understand that the functional assay was performed to demonstrate that the protein was in its native functional state but the lack of purity of the sample raises doubt regarding the results.

Our response:

We thank for this insightful comment. The concern is indeed valid and we have accordingly performed a series of control experiments to address this issue. Taken together, they support that our observations are pertinent to intact and fully functional SERT. We believe that these new experiments have considerably strengthened our findings.

Firstly, we have now performed SEC analysis of purified SERT. This showed three main peaks, one corresponding to the void volume of the column, one corresponding roughly to monomeric SERT, and finally one corresponding to the elution of imidazole. SDS-PAGE from the two first peaks showed that the majority of SERT was in the peak corresponding to monomeric SERT. We note, however, that SERT is glycosylated, adding a level of native heterogeneity and the sample was solubilized in detergent that results in micelles of varied sizes and thus SEC is in fact not the ideal tool to assess for partial unfolding or aggregation of SERT. In accordance, SDS-PAGE revealed that smaller amounts of SERT

were also detected in fractions collected between the main SERT peak up to and including the void volume peak. To investigate the conformational properties of SERT from the monomeric peak containing the majority of SERT, fractions were collected and up-concentrated and subjected to HDX-MS analysis across three labeling time points (0.25 min, 10 min, 60 min). Importantly, the HDX profile of this SEC-purified SERT was nearly identical similar to that reported in the manuscript on SERT samples that had not been subjected to prior SEC-purification. Only exceptions were two peptides spanning residues 89-94 and 89-95 when measured at 10 min and 60 min. However, this could be explained by an observed increase in carry-over for the two peptides in the latest HDX-MS experiment. These locally-resolved HDX-MS measurements are very sensitive to detect even modest local changes in SERT conformation between samples and we can thus confidently conclude that the HDX-MS experiments we report in the manuscript are pertinent to a non-aggregated detergent:micellar solubilized state of functional SERT. Furthermore, as also discussed in our reply to Comment #1 by Reviewer 1 we can also confirm that SERT samples used for HDX-MS in the manuscript furthermore did not contain any measureable degradation products as assessed by western blotting (see our reply to Comment #1 by Reviewer 1).

Secondly, to further address the stability of the purified SERT sample, we performed control HDX-MS experiments on purified SERT samples that were incubated at variable times at 4°C (up to 1 hour) and subsequently 30 min at 25°C. In both cases, SERT yielded similar HDX profiles, suggesting that the incubated time period did not perturb SERT structure. Similar control HDX-MS experiments were performed for SEC-purified SERT samples. Again, HDX profiles were almost identical. Accordingly, we are confident that SERT remains folded with no aggregation induced during the actual labeling step of the experiment. Circular dichroism and FTIR analysis that provide only a global conformation would not be nearly as sensitive to detect structural changes and furthermore the requirement for salts, detergents and lipids to keep the SERT sample stable which would interfere with such analyses e.g. CD analysis highly influenced by the presence of chloride ions which is required for SERT stability.

The data and explanatory text for these extensive additional experiments have now been added to the Methods (p. 20) of the main manuscript and as a chapter in the supplementary information "Evaluating the purity and stability of hSERT" and as Supplementary Figure 1b (western blot of SERT) and Supplementary Figure 2 (HDX-MS butterfly plots comparing the HDX of the SERT following the different sample treatments).

Related to the questions concerning the fast HDX of some parts of TM in SERT:

The only regions of SERT in which we observe high levels of HDX after 15 sec (Figure 1) are in TMs 2, 3, 6, and 10.

We actually observed a similar behavior in the corresponding helices of DAT (unpublished results) and LeuT in both detergent and nanodiscs (DOI: 10.1126/sciadv.aar6179) and so we believe this is a conserved feature of the dynamics of NSS proteins. The TM helices 2, 3, 6, and 10, are partially solvated and would thus have ready access to solvent. Furthermore, the peptide that we use to probe the HDX of TM2 included also parts of the nearby loops which could, in this case, account for the fast deuterium uptake.

To clarify on the comment of the reviewer we have now added the following text to the manuscript on p. 8 along with a Supplementary Figure:

“We note that the observed flexibility of parts of TM3, TM6 and TM10 could be due to local structural perturbations caused by the non-native solubilization conditions employed (DDM detergent micelle). However, the observed full binding activity of detergent-solubilized SERT used for HDX-MS makes this unlikely and furthermore the X-ray crystal structure of SERT reveals these helices to be partially solvated, which provide an alternate explanation for their elevated HDX. To address whether HDX in some SERT domains could be influenced by partial unfolding, we performed HDX-MS experiments on purified SERT batches after sample pretreatment steps that involved storage at room temperature for 1 hour. All samples yielded similar HDX profiles (Supplementary Fig. 2). Such locally-resolved measurements are very sensitive to detect even modest local changes in SERT conformation due to partial unfolding or aggregation during the time-frame of the HDX experiment.”

Comment #4:

The presence of a mix of lipids throughout the whole experiment is intriguing. Could the authors explain why they add lipids? Is it a common practice for hSERT purification? Does it contribute to the stability of the protein? The presence of lipids is not trivial because it has been shown to modulate the conformational dynamics of other transporters and receptors, even in the absence of a bilayer (doi: 10.1038/nchembio.1960, doi: 10.1038/nsmb.3262). The authors might want to justify the use of lipids and discuss this further.

Our response: Lipids and cholesterol are important components in the function and stability of eukaryotic membrane proteins. This particular lipid:detergent:cholesterol composition is the same as Coleman et al. used when they reported the first crystal structure of hSERT in April 2016. Both cholesterol and lipids were observed to interact with the transporter in the structure, probably by stabilizing specific domains. Previously, we have been using the cholesterol mimicking detergent, digitonin, with some success, but the lipid mix is superior. However, upon changing to the current lipid:detergent:cholesterol mix, we observed improvements in the stability and activity of our purified SERT samples. We have now added a sentence and a reference in the methods section p. 18 justifying the use of lipids during sample preparation.

“The detergent and lipids added during SERT purification have been previously reported along with the structure of hSERT by Coleman et al.⁸.”

Comment #5:

A recent study by Anderlüh A, et al, Nat Commun. 8, 14089, pointed to the formation of SERT oligomers. The study suggested the arrangement allows for formation of both dimers and other higher order oligomers. This oligomeric formation likely to be mediated by lipids. However, molecular level details of the interaction between the lipid and SERT have yet to be elucidated. Do the authors have any evidences of oligomeric formation? Or are they certain of the monomeric arrangement for SERT? If SERT oligomerises, the presence of more than one monomers may introduce asymmetry in the reported HDX data. The authors may want to discuss this issue in their paper.

Our response: We thank the reviewer for this insightful comment. We do not see any asymmetry in the reported HDX data on SERT in any of the states examined. Furthermore, the new SEC analyses performed do not indicate significant dimerization in the SERT samples used for HDX-MS (see our reply to comment 3 above). We therefore do not believe that dimerization is prevalent at the low concentration used during the HDX experiments.

Other concerns of Reviewer 2

Concern #1:

What is the meaning of “ligand-attenuated changes”?

Our response: The meaning of “ligand-attenuated changes” is the same/similar to “ligand-mediated changes”. We have now changed the wording throughout to avoid confusion.

Introduction

Concern #2:

There is a lack of references overall, e.g.: crystal structure of LeuT, pharmacological uses of SSRIs, etc. Please acknowledge previous work.

Our response: We welcome the opportunity to make more references and did not do so initially in an attempt to confine to journal guidelines. We do reference the crystal structure of LeuT, albeit at the end of the paragraph in which we discuss it (ref 6, p.4). Likewise for the crystal structures of DAT and SERT.

To accommodate the reviewer, we now also include references to LeuT, SERT and DAT structures immediately when first mentioned (line 46 p.3 and line 81 p.5) as well as references to pharmacological uses of SSRIs on line 41 on p.3.

Concern #3:

Line 77: “a role of K⁺ has been shown for SERT function”. Be specific.

Our response: We thank the reviewer for this comment. We have now rewritten the sentence and included it here:

“A role of K⁺ has been shown for SERT function, that is K⁺ binds to the inward-facing transporter and is transported out of the cell, rendering the transport charge-neutral²² and speeding up the return-step of the transport cycle²³.”

Concern #4:

Line 80: “functional” conservation should be replaced by “mechanistic” – NSS transport different substrates so they don’t have the same function – but they might do it in a similar way.

Our response: We thank the reviewer for this comment and have now changed the wording throughout the manuscript.

Results

Concern #5:

I’m intrigued by the use of rhizopuspepsin. Could the author mention how they obtained that type of pepsin? What benefits does it offer?

Our response: We thank the reviewer for this inquiry which also brought our attention that an acknowledgement of its source is missing. Rhizopuspepsin (also referred to as Protease type XVIII) was kindly provided by Petr Man, a collaborator at BioCev, Czech Academy of Sciences. Despite being called rhizopuspepsin, it is in fact not a variant of pepsin. Like pepsin, rhizopuspepsin has broad cleavage specificity but it differs from that of pepsin. In a paper by Rey et al. (2009; DOI: 10.1002/rcm.4260) where P. Man was a co-author, immobilized rhizopuspepsin was reported as a tool for HDX-MS where they discuss cleavage preference shortly. We have now made reference to this paper on page 7 and made an acknowledgement on page 23.

Concern #6:

The authors use a 95 confidence interval that is not very stringent. The authors might consider using a 98 or even 99 CI and see whether they have more consistency between the region reporting conformational changes in the different experiments. For example, in figure 2, the authors include on the figure the peptide 483-499, but looking at the uptake plot, the difference seems quite small and it might be insignificant. The authors actually seem to share this opinion since they don't mention the peptide in the main text. In the same paragraph, the authors state that the overall uptake pattern is consistent with an intracellular opening coupled to an extracellular closing upon K⁺ binding. This doesn't seem a statement representing the data on fig.1 as we see uptake on both sides, but using a more stringent confidence interval might prove them right.

Our response: We thank the reviewer for the insightful suggestion. The choice of confidence interval was carefully selected to provide a general cut-off value, based on a average standard deviation for all data points measured in triplicates, above which we consider differences in HDX to be significant. However, as the reviewer will no doubt be aware the actual error is not the same for each peptide and each data point. Thus a students t-test can provide a more reliable estimator of significance in each peptide. We have now reconsidered the threshold value, a peptide has to cross to be considered significant. The new requirement is stated on page 22.

“In the comparative HDX analysis of the different functional states, a peptide was only considered to have a significant difference in HDX if two consecutive time points showed a significant difference in deuterium incorporation above the average CI or based on a Student T-test ($p < 0.01$). As an exemption to this rule, a difference was also considered significant if the last time point (480 min) displayed a difference in HDX at least two times larger than the CI, as this indicates a significantly expanding difference not fully sampled by the HDX time course.”

Importantly, the adoption of this more conservative significance threshold does not change the major conclusions of the manuscript. However, some of the minor differences in HDX between Na⁺ reference state and the K⁺ and 5-HT states within segments 126-150, 217-235, 319-329 (effect lost in 5-HT state only), 483-499, 616-634, 623-641, are no longer significant according to this new threshold and the text in the results and discussion of the manuscript were thus modified accordingly.

Concern #7:

Line 175. I'm not sure I follow the line of thoughts of the authors when they claim that the observed EX1 kinetics on TM1b correlates with a closing of the extracellular side. Could they be more explicit and explain this further? Overall there is more uptake in the presence of K⁺.

Our response: The selective induction of EX1 kinetics in TM1b of SERT upon binding K^+ shows that this important part of the protein undergoes much slower structural refolding rates in the K^+ -state than in the other states examined. As our work indicates that conformational changes in TM1b is a hallmark event in the opening/closing of the extracellular side (along with other parts) we therefore propose that the opening and closing motions of this part of the extracellular gate is slowed by the binding of K^+ .

We agree that this can be clarified further in the text to avoid misunderstandings. We have now changed the sentence on Line 175 (lines 187-191 in revised version) as follows:

“It suggests that the dynamics of the extracellular side of TM1 is considerably slowed by the presence of K^+ . TM1b forms part of the extracellular gate and is known to be involved in substrate transport. By extension, our results suggest that the opening and closing motions of the extracellular gate is significantly modulated and slowed by the binding of K^+ relative to the Na^+ -bound SERT.”

Concern #8:

Line 178: Why do the authors control the role of imidazole on HDX if they exchange buffer overnight by dialysis? There shouldn't be any imidazole around anymore.

Our response: We agree that dialysis removes imidazole for the SERT sample. We used SERT that was buffer exchanged overnight by dialysis in the majority of the reported HDX experiments. However due to the considerable consumption of SERT sample in the ligand-binding HDX experiments (i.e. S-citalopram, cocaine and ibogaine) we did not perform dialysis on the SERT used for these studies, thus mitigating the considerable sample loss during this procedure. Instead we lowered the imidazole concentration to 36 μ M by diluted the sample 1:4 (as described in the Methods section). We therefore performed the described control experiment to ensure that our results from binding experiments were not interfered by the presence of imidazole. Importantly, the control experiment showed that while imidazole had a conformational impact on SERT in the presence of K^+ , no impact was detected on the conformation of SERT in the presence of Na^+ which was used as the reference conditions for all ligand binding experiments.

Concern #9:

The authors observe important HDX in regions that are supposedly α -helices and are expected to undergo slow HDX. Can the authors ensure that the protein is folded?

Our response: Binding experiments on the purified SERT used for HDX-MS showed the expected binding profile of active SERT reported elsewhere (see Line 124, Page 6). We believe this complimentary experiment strongly supports that the SERT used has a correct folded structure and active site and is fully capable of binding substrate.

Furthermore, our HDX-MS measurements on SERT correlate overall with the expected secondary structure of folded SERT. For further details, we refer to our reply to a similar comment made by Reviewer #1 (comment #3). We include the reply here for convenience:

The only regions of SERT in which we observe high levels of HDX after 15 sec (Figure 1) are in TMs 2, 3, 6, and 10.

We actually observed a similar behavior in the corresponding helices of DAT (unpublished results) and LeuT in both detergent and nanodiscs (DOI: 10.1126/sciadv.aar6179) and so we believe this is a conserved feature of the dynamics of NSS proteins. The TM helices 2, 3, 6, and 10, are partially solvated and would thus have ready access to solvent. Furthermore, the peptide that we use to probe the HDX of TM2 included also parts of the nearby loops which could, in this case, account for the fast deuterium uptake.

To clarify on the comment of the reviewer we have now added the following text to the manuscript on p. 8 along with a Supplementary Figure:

“We note that the observed flexibility of parts of TM3, TM6 and TM10 could be due to local structural perturbations caused by the non-native solubilization conditions employed (DDM detergent micelle). However, the observed full binding activity of detergent-solubilized SERT used for HDX-MS makes this unlikely and furthermore the X-ray crystal structure of SERT reveals these helices to be partially solvated, which provide an alternate explanation for their elevated HDX. To address whether HDX in some SERT domains could be influenced by partly unfolding, we performed HDX-MS experiments on purified SERT batches after sample pretreatment steps that involved storage at room temperature for 1 hour. All samples yielded similar HDX profiles (Supplementary Fig. 2). Such locally-resolved measurements are very sensitive to detect even modest local changes in SERT conformation due to partial unfolding or aggregation during the time-frame of the HDX experiment.”

Concern #10:

It would be helpful to annotate the topological maps on the figures with helices number etc.

Our response: We thank the reviewer for this comment and we have now performed the suggested annotation on all figures.

Concern #11:

Why is peptide 217-235 coloured in fig.3? it doesn't seem that anything is going on there when looking at the uptake plots. Again, there is no mention of this peptide in the main text anyway.

Our response: We thank the reviewer very much for pointing this out. This is a mistake during the coloring of effects of 5-HT on SERT and we also do not discuss this in the main text anyway. We have now corrected this in a revised version of Figure 3.

Concern #12:

Lines 198-199: It sounds like an over-claim that changes induced by 5-HT probably indicate the inward-facing state. It appears to me that mixed signals are obtained from HDX-MS that are difficult to attribute to a specific state. The authors should modify this statement accordingly.

Our response: We agree with the reviewer that HDX-MS cannot conclusively infer that 5-HT induces an inward-facing state. We therefore carefully phrased this section, specifically use “may” to leave

interpretation open and only focus on the extracellular gate and not a specific complete inward-facing state. We have now modified the section (p. 10) to accommodate the concern of the reviewer.

“The data presented here clearly suggest a difference in the substrate-bound conformations of LeuT and hSERT: Whereas LeuT is further stabilized in its Na⁺-bound state by leucine^{13,18}, 5-HT binding to hSERT both stabilizes but also destabilizes regions relative to the Na⁺-bound form. hSERT has been proposed to favor an inward-facing conformation in the presence of 5-HT⁴⁸, and the structural stabilization observed in TM1b by 5-HT may be indicative of a closing of the extracellular gate. Our HDX data however does not support a complete transition to an inward-facing state similar to suggestions in previously reported MD simulations²⁶.”

Concern #13:

Line 200: I'm not sure it makes sense to compare the K⁺ state with the Na⁺/5-HT state.

Our response: We agree and for the same reason, the comparison is not done directly between the K⁺ state and the Na⁺/5-HT state, but rather we qualitatively compare differences between Na⁺ and K⁺ on the one hand and Na⁺ and Na⁺/5-HT state on the other.

Discussion

Concern #14:

Line 273: Replace “native” by “in-solution”. The absence of a bilayer, a gradient and whatever other components that are part of an intact cell preclude the use of that word.

Our response: We agree and have now changed through-out the manuscript.

Concern #15:

Line 284: Upon K⁺ binding.

Our response: We thank the reviewer for this comment and have now added the missing word.

Concern #16:

Line 301: I question the significance of the difference observed for the C-terminal peptides. What happens if you use a higher confidence interval?

Our response: To accommodate the request of the reviewer, we have now changed the threshold for significance used in the manuscript, please see our reply to a similar comment made by the same reviewer (concern #6). With the exception of a peptide including part of TM12 (residues 595-607 further isolated to residues 596-598), no segments with significant differences in HDX according to this new threshold now remain in the C-terminus.

Concern #17:

Line 315: “these regions did not show a significant difference in HDX”. Obvious question but did these regions displayed any HDX at all? If they are too protected, by the micelle for example, I'm not surprised.

Our response: We thank the reviewer for this question. The reviewer correctly points out that these segments have very limited or no HDX. To this, a segment spanning residues 483-499 (including Thr479) was an exception, being around or over 80% deuterated at the longest HDX labeling time point.

Concern #18:

Line 343: I agree that the HDX behaviour of TM1b seems to be a hallmark of the conformational a transition and I would suggest presenting it in a cartoon.

Our response: We thank the reviewer for this comment. We refer to our reply to comment #2 from the same reviewer where we have included additional text to the paragraph (line 343 (p.16, line 353-373 in revised version)) and a new figure (Figure 5).

Figures

Concern #19:

The figures are overall of high quality and clear to follow. What is somewhat counterintuitive is the presentation in Figures 2A, 3A and 4A-C. I believe that the authors plot the Delta HDX of (Na⁺ - ion/substrate/inhibitor state). This is opposite of what it is plotted in panes 2B, 3B and 4D-F. It will therefore be more consistent if they will plot Delta HDX of (ion/substrate/inhibitor state - Na⁺ state).

Our response: We thank the reviewer for this comment and agree that this could help non-expert readers. We have now rephrased the figure text to make it the color coding independent of the delta HDX.

“Regions showing significant increase (red) or decrease (blue) in dynamics (i.e. HDX) in the presence of...”

Concern #20:

I am surprised by the statement that the protein was deemed pure enough. The gel doesn't show a pure protein. This raises questions about the results of the binding assay and the overall stability of the protein.

Our response: We would like to refer to our reply to a similar comment by Reviewer 3, comment #1.

REVIEWER 3

Comment #1:

The authors present an SDS-PAGE gel to illustrate the hSERT sample used for subsequent analysis. The protein sample is highly contaminated with other species. On the one hand, this is an excellent demonstration of the robustness and sensitivity of HDX-MS. However, I recommend a few cautionary notes regarding sample homogeneity:

- a. There are several mentions of “purified” protein, yet the gel indicates that “enrichment/enriched” are more accurate terms.
- b. The authors note regulatory interactions with SERT (line 306). Given that the SERT protein was expressed and purified from a non-neuronal, recombinant source (i.e., an embryonic kidney cell line), I am largely satisfied that

the contaminating proteins are unlikely to interfere or interact with SERT. However, it would be worth pointing out this caveat explicitly in the text somewhere.

c. Notably, contamination with SERT proteolysis products (from endogenous proteases liberated during cell lysis) is a potential risk in non-homogenous protein samples. The HDX data does not seem consistent with significant contamination with proteolyzed SERT. Nevertheless, the authors could clarify the sample quality by Western blotting with a polyclonal antibody for hSERT or an anti-His tag antibody.

Our response: We thank the reviewer for this comment and the helpful suggestions.

- a) We agree that the purified hSERT is not pure and but due to selectivity and sensitivity of the HDX-MS it does not need to be. We believe that the term purified is accurate as this term does not mean pure. We have now added an estimation of the purity of the sample (20-25%) as based on the measured intensity of the bands in a SDS-PAGE, measured at 700 nm.
- b) We thank the reviewer for suggesting to state that hSERT was expressed from a non-neuronal cell line. We have now added the following sentence in the Results part (p.7, line 126-128:

“As SERT was expressed and purified from a non-neuronal, recombinant source (i.e., an embryonic kidney cell line), contaminating proteins are unlikely to interfere or interact specifically with SERT.”

- c) We have now performed a western blotting analysis of the purified hSERT (natively glycosylated) used for HDX-MS using an antibody specific for the C-terminal end of hSERT. The results clearly show the absence of detectable SERT proteolysis products and thus demonstrate the relative purity of the sample with regards to SERT related contaminants. We have now included this data in the Supporting Information and refer to this new experimental data on page 6 in the manuscript text and on and in the supplementary information:

“...the absence of SERT proteolysis products in the hSERT sample used for HDX-MS was confirmed by western blotting analysis (Supplementary Fig. 1).”

Comment #2:

The HDX was performed on DDM-detergent-solubilized SERT. This approach appears to be standard in the field, and the authors include binding data demonstrating that the protein retains function with DDM. However, micellar detergent “belts” may influence structure differently than the native planar bilayer. Generally, this variable is isolated, as the authors use the Na⁺-bound, DDM-solubilized sample as a baseline control. There is one location in the Results where this potential difference between solubilizing micellar lipids vs. planar bilayer should be addressed. The paragraph describing intrinsic dynamics (lines 140-145) notes unusually fast exchange rates in TM helices evident in the crystal structure. The authors should include the non-native solubilization conditions as a possible alternative explanation for these exchange rates.

Our response: We do not deem it likely for reasons described in our reply to a similar comment by Reviewer 2 (Concern #9).

We actually observed a similar behavior in the corresponding helices of DAT (unpublished results) and LeuT in both detergent and nanodiscs (DOI: 10.1126/sciadv.aar6179) and so we believe this is a conserved feature of the dynamics of NSS proteins. The TM helices 2, 3, 6, and 10, are partially solvated and would thus have ready access to solvent. Furthermore, the peptide that we use to probe

the HDX of TM2 included also parts of the nearby loops which could, in this case, account for the fast deuterium uptake.

To clarify on the comment of the reviewer we have now added the following text to the manuscript on p. 8 along with a Supplementary Figure:

“We note that the observed flexibility of parts of TM3, TM6 and TM10 could be due to local structural perturbations caused by the non-native solubilization conditions employed (DDM detergent micelle). However, the observed full binding activity of detergent-solubilized SERT used for HDX-MS makes this unlikely and furthermore the X-ray crystal structure of SERT reveals these helices to be partially solvated, which provide an alternate explanation for their elevated HDX. To address whether HDX in some SERT domains could be influenced by partly unfolding, we performed HDX-MS experiments on purified SERT batches after sample pretreatment steps that involved storage at room temperature for 1 hour. All samples yielded similar HDX profiles (Supplementary Fig. 2). Such locally-resolved measurements are very sensitive to detect even modest local changes in SERT conformation due to partial unfolding or aggregation during the time-frame of the HDX experiment.”

Comment #3:

Figs 1-4 share a view of the SERT structural model, upon which the exchange perturbations are mapped. It may clarify the topology of the protein in the membrane to annotate Fig. 1 with labels for the “extracellular” and “intracellular” sides. Consider also an estimation of the bilayer position with dotted lines denoting the lipid headgroup layers.

Our response: We thank the reviewer for this comment and have now revised Figs 1-4 to indicate both extracellular and intracellular sides and an estimation of the lipid bilayer position.

Comment #4:

On lines 193-194, the authors point out a functionally important salt bridge informed on by the HDX results. A small figure (an inset or SI figure, perhaps) illustrating the side chain interaction with the HDX data mapped would be helpful.

Our response: We thank the reviewer for this comment and have now also indicated the proposed Arg79-Asp452 saltbridge in a new figure that highlight the changes in HDX observed in the key TM1 region (see also the suggestion by Reviewer 2, concern #18).

Comment #5:

There is a brief interlude (lines 178-184) wherein the authors compare whether the presence of imidazole influence exchange dynamics in the presence of Na⁺ vs. K⁺. I'm not entirely clear on why this is included in the manuscript, and a brief note of justification beyond “as a control” would be helpful. Perhaps it could be relegated to the Supporting Info? Minimally, the concentration of imidazole used should be included in the figure (SI Fig. 4).

Our response: We welcome the opportunity to clarify. We used affinity-purified SERT that was buffer exchanged overnight by dialysis in the majority of the reported HDX experiments. However due to the considerable consumption of sample in the ligand binding HDX experiments (i.e. S-citalopram, cocaine

and ibogaine) we did not perform dialysis on the SERT used for these studies, thus mitigating the considerable sample loss during this procedure. Instead we lowered the imidazole concentration to 36 mM by diluting the sample 1:4 (as described in the Methods section). We therefore performed the described control experiment to ensure that our results from binding experiments were not interfered by the presence of imidazole. Importantly, the control experiment showed that while imidazole had a (minor) conformational impact on SERT in the presence of K^+ , no impact was detected on the conformation of SERT in the presence of Na^+ which was used as the reference conditions for all ligand binding experiments.

We have now included an explanatory sentence on page 18 and included the concentration of imidazole in SI Figure 4:

“purified hSERT was dialyzed or diluted 1:4 (in the case of the HDX-MS experiments with inhibitors)

Comment #6:

Supplementary Fig. 1 includes a BSA standard curve in the SDS-PAGE gel. Given the use of gel staining for protein quantification, the staining technique used should be stated in the figure legend.

Our response: We thank the reviewer for pointing this out and have now added this information to the figure legend.

Comment #7:

Supplementary Fig. 4: I am unclear about the meaning of “summed deuterium uptake” to describe the gray bars. Can the authors clarify?

Our response: We thank the reviewer for this inquiry. The summed deuterium uptake should in fact read “the sum of the difference in HDX” i.e. the sum of the difference in deuterium uptake between the two states. However, for clarity and in keeping with the format of all the other uptake plots, we have now removed the gray bars and revised the figure and figure legend for SI Figure 4.

Comment #3:

A few typos: line 265, “outmost”?; line 130 “...and at least other 12% form...”

Our response: We thank the reviewer for pointing these out – we have now corrected them in the revised version.

Reviewers' Comments:

Reviewer #1:

Remarks to the Author:

The authors have responded to points of critique and suggestions

A significant issue raised was on a poor purity of the protein, and unknown conditions of protein stability and functional state; factors that all affect the obtained results and therefore the interpretations and strength of conclusions.

Re. purity (author response to reviewer 1, comment 1): The authors do not provide a good response here - rather than engaging in solutions for quite manageable steps to improve purity, they respond through questionable explanations on how membrane proteins cannot be purified well (response to reviewer), and that coming from an expression system it will not contain interacting contaminants (note added in the revised manuscript). These statements are not valid. His-tag is not very efficient for small yields, no matter the origin of the protein – Mammalian membrane proteins can certainly be purified at high level, it only depends on the expression level, affinity tag and protocol used. Secondly, SERT is expressed in many different tissues and may certainly pick up contaminating interaction partners or adhering proteins from HEK293 cells, and at the low level of purity presented here it must be considered with proper disclaimers, and not ruled out by invalid arguments.

comment 2

the authors do not respond to the comment that the low purity leads to a poor reproducibility of ligand occupancy and therefore state-to-state comparisons interpreted as dynamics.

comment 6 (and reviewer 2 comment 1 and 2)

the authors do not comment much on MD simulations - the paper needs a better introduction and final discussion of what structures and dynamical studies have shown earlier, and how the observed dynamics of this study compare (well) to previously determined structures of SERT and SERT homologs. The discussion can at the same time be shortened. What the comparative narrative will show is that the results presented here fit very well to expectations - the findings of the paper are largely confirmatory

The difference between SERT-5HT and LeuT-Leu makes excellent sense, comparing a transporting, active complex (SERT) to an almost inactive, trapped transporter complex crystallized in a blocked outward-oriented conformation. Inhibitor-bound complexes are stabilized in the conformation as expected, and the ibogain also confirms earlier findings of a different site stabilizing an inward-facing state

Overall, the authors should change the manuscript to a more descriptive/confirmatory record of achievements - and it will still be a very noble contribution like that

Reviewer #2:

Remarks to the Author:

The authors have made an excellent job reviewing the article. It will make an exciting contribution to the field and I have no hesitation to recommend it for publication in Nature Communications.

REVIEWER 1

Comment #1:

The protein is not very pure (suppl. fig. 1). Did the authors try a cleavage of the C-terminal tag (seemingly not) followed by another round of reverse Ni²⁺ purification, i.e. capturing all impurities again, but not the released SERT?

Our response (1st revision): We thank the reviewer for the opportunity to clarify. Mammalian NSS proteins, like SERT, are notoriously difficult to express and purify and the suggested procedure was not attempted. The reasoning was that the transporter exhibited the expected binding profile ($K_D = 7.5 \pm 0.7$ nM; Supplementary Figure 1c) from the literature (Talvenheimo et al. (1983; PMID: 6853478); DOI: 10.1021/bi048022b) without removing the C-terminal tag and the extra steps would lead to unwanted loss of protein and thus dilution of the protein sample. This would have proven problematic to the subsequent HDX-MS experiments as the protein was already at quite low concentration following the first Ni²⁺-NTA affinity purification.

We have performed additional control experiments to assess the nature of impurities in the SERT sample and can confirm that these are not SERT degradation products (please see our response below to Comment #2 of the same reviewer for details).

Reviewers response: The authors have responded to points of critique and suggestions. A significant issue raised was on a poor purity of the protein, and unknown conditions of protein stability and functional state; factors that all affect the obtained results and therefore the interpretations and strength of conclusions.

Re. purity (author response to reviewer 1, comment 1): The authors do not provide a good response here - rather than engaging in solutions for quite manageable steps to improve purity, they respond through questionable explanations on how membrane proteins cannot be purified well (response to reviewer), and that coming from an expression system it will not contain interacting contaminants (note added in the revised manuscript). These statements are not valid. His-tag is not very efficient for small yields, no matter the origin of the protein – Mammalian membrane proteins can certainly be purified at high level, it only depends on the expression level, affinity tag and protocol used. Secondly, SERT is expressed in many different tissues and may certainly pick up contaminating interaction partners or adhering proteins from HEK293 cells, and at the low level of purity presented here it must be considered with proper disclaimers, and not ruled out by invalid arguments.

Our response (2nd revision): We welcome the opportunity to clarify. We would like to emphasize that as part of the last revision we did perform a control HDX experiment of SERT at significantly higher purity after SEC purification. We included this data in Supplementary Fig. 2 and SEC-purified SERT shows nearly identical HDX (within experimental error) to SERT without SEC-purification. As for the specific concern about the impact of the presence of impurities during HDX ligand-binding experiments (Comment #2), we have now added the requested disclaimer in the Methods section for Comment #1 and parts of Comment #2:

“We note that the HDX-MS ligand-binding experiments were performed using ligand concentrations which should ensure ~90% binding occupancy of hSERT, excluding effects caused by differences in affinity by any of the ligands. However, as the transporter was

around 20-25% pure during HDX-MS (Supplementary Fig. 1), variability could occur in the binding occupancy of hSERT if the ligands displayed different unspecific binding to protein impurities present. Such a scenario would impact the exact magnitude of HDX differences observed, thus while our results reveal which regions of hSERT are perturbed by different ligands, a detailed quantitative comparison of HDX differences between different ligands should be approached with caution. Further, though purified from a non-neuronal cell line, some hSERT interacting proteins could remain in the sample. These possible interaction partners would however be present in all states (including the reference state) and any impact on hSERT should thus be the same in all HDX experiments. Additionally, in a control experiment where the HDX of the Na⁺-bound reference state was performed following size exclusion chromatography (SEC), and thus under conditions with significantly fewer impurities, hSERT showed identical HDX, within the experimental error, to affinity-purified hSERT (Supplementary Fig. 2)."

We have now also made a specific reference to this disclaimer in the Discussion on page 15).

Comment #2:

The impurities among other concerns may affect the accurate titration at 90% occupancy and therefore state-to-state comparisons. Even though e.g. S-citalopram and 5-HT binding is supposedly specific and of high and medium-high affinity, respectively (as determined by the SPA assay) it may still be flawed by unspecific binding. It depends on which background control samples were used for subtraction of unspecific binding. SERT-free membranes would be the right control, but this is not stated in the methods.

Our response (1st revision): We welcome the change to clarify. Some unspecific binding is usually present in radioligand binding assays and a control for unspecific binding was performed using high concentrations of another high affinity ligand, in this case paroxetine ([Paroxetine] = 10 μM; K_D = 0.13 nM (DOI: 10.1016/S0014-2999(97)01393-9)). With the ligands sharing the same binding site and with a high concentration of the higher affinity, non-radiolabeled paroxetine, we are convinced that any residual signal can be used to accurately quantify the unspecific binding of [³H]-imipramine. We believe this to be standard method and superior for background subtraction relative to performing a purification procedure of SERT-free membranes. The over-expression of SERT causes a considerable change in cell composition and hence the membranes will not be comparable.

It is stated in the Methods section on p. 20:

"Unspecific binding (10 μM paroxetine) was determined in parallel... Binding activity was quantified on a MicroBeta scintillation counter (PerkinElmer). Following subtraction of the counts caused by non-specific binding..."

Furthermore, we have now directly investigated the occurrence of SERT degradation products in the SERT sample used for HDX-MS and western blotting analysis reveal these to be negligible.

This additional data has now been included in the Supporting Information and are referred to in the main text on p. 7:

“SDS-PAGE analysis of the purification indicated around 20-25% purity and the absence of SERT proteolysis products in the hSERT sample used for HDX-MS was confirmed by western blotting analysis (Supplementary Fig. 1).”

Reviewers response: The authors do not respond to the comment that the low purity leads to a poor reproducibility of ligand occupancy and therefore state-to-state comparisons interpreted as dynamics.

Our response (2nd revision): We thank the reviewer again for his comment. During the first revision, we understood the comment as being directed towards the activity testing of the transporter (i.e. the SPA assay) rather than the binding occupancy during the HDX reactions.

The binding occupancy was calculated using K_d of the individual substrate and inhibitor, and the concentration of the transporter as determined by SDS-PAGE. The reviewer correctly points out that due to the low purity of the transporter, the calculated 90% binding occupancy may not be accurate and may fluctuate depending on how the protein concentration is determined. Due to the excess of ligand present, these inaccuracies could have an impact on the magnitude of HDX differences observed upon binding two different ligands mainly if the two ligands also had different unspecific binding to impurities copurified with SERT. We do not deem this likely, as there were no SERT-degradation impurities present (see our western blotting analysis, Supplementary Fig. 1) but we understand the point the reviewer is making and that we cannot exclude this. We would however like to emphasize that we do not make extensive quantitative comparisons of the magnitude of differences in HDX between ligand-bound states of SERT. The main part of the manuscript where this is done is in a brief comparison of the magnitude of HDX differences between the ibogaine-state and the potassium-state (page 16). To accommodate the reviewers concern we have now removed these sentences from the manuscript.

Further, we have now added to the statement of how we calculate the binding occupancy in the method section in hope to accommodate the reviewers concern regarding the possible impact of protein impurities on the calculated binding occupancy (lines 428-432):

“5-HT or inhibitors were added in a concentration that yielded a 90% binding occupancy of hSERT (0.31 μ M) during the deuterium labeling reactions, based on published affinities (ligand concentrations: 10 μ M, 0.34 μ M, 2.4 μ M, and 25.5 μ M, for 5-HT (K_d : \sim 1 μ M)^{45,64}, S-citalopram (K_d : \sim 5 nM)^{5,45,65}, cocaine (K_d : \sim 220 nM)^{5,45}, and ibogaine (K_d : \sim 2.6 μ M)^{56,66}, respectively).

Finally, we have also added the requested disclaimer in the Methods section concerning the possible impact of SERT purity on the HDX ligand binding experiments (see our additional reply to comment #1):

Comment #6:

Dynamics have been extensively studied with MD simulations in many groups, and there are also many studies probing SERT dynamics and conformations with reagents. The sentence “current knowledge about the conformational dynamics of SERT is limited and is based on inferences from biochemical and mutagenesis studies23-25” (lines 88-89) therefore seems inappropriate.

Our response (1st revision): The reviewer is correct that the molecular dynamics of LeuT have been studied by MD simulations in several groups. However we do believe that our knowledge from experiments of the conformational dynamics of SERT is quite incomplete and largely based on biochemical probes and mutagenesis studies that require covalent modification of the protein which can introduce non-native artefacts.

To accommodate the request, we have now rephrased the relevant section on page 5 as follows:

“However, in spite of the apparent structural similarity between LeuT and SERT, and several MD simulations of SERT²⁵⁻²⁷, our knowledge from experimental data on the conformational dynamics of SERT is incomplete and based on indirect inferences from biochemical and electrophysiological studies²⁸⁻³² often using SERT mutants.”

Reviewers response: The authors do not comment much on MD simulations - the paper needs a better introduction and final discussion of what structures and dynamical studies have shown earlier, and how the observed dynamics of this study compare (well) to previously determined structures of SERT and SERT homologs. The discussion can at the same time be shortened. What the comparative narrative will show is that the results presented here fit very well to expectations - the findings of the paper are largely confirmatory

The difference between SERT-5HT and LeuT-Leu makes excellent sense, comparing a transporting, active complex (SERT) to an almost inactive, trapped transporter complex crystallized in a blocked outward-oriented conformation. Inhibitor-bound complexes are stabilized in the conformation as expected, and the ibogain also confirms earlier findings of a different site stabilizing an inward-facing state

Overall, the authors should change the manuscript to a more descriptive/confirmatory record of achievements - and it will still be a very noble contribution like that

Our response (2nd revision): We agree that our response to this question was not comprehensive. We have largely compared our results to evidence based on x-ray crystallography and from the experiments on conformational dynamics in LeuT.

We have now substantiated our observations with several MD simulations. Accordingly, our observations from 5-HT binding are supported by the following in the Discussion (line 317-322):

“Similar results have been obtained with unbiased MD simulations of SERT in complex with Na⁺ and 5-HT⁶⁰. They found that 5-HT binding leads to an isomerization of SERT to an inward-facing state, largely mediated by conformational changes in TM1a and 6b. Also in accordance with the present results, they propose that the two halves of these helices are internally flexible with the intracellular part of the helices moving considerably more than the extracellular part.”

In addition, we have added further details and references to the description of S-citalopram binding. The sentence now reads (line 326-329):

“According to the hSERT crystal structure⁹ and several MD simulations^{25,61–63}, S-citalopram is suggested to interact with both Tyr95 and Asp98 in TM1 but also with other SERT regions, including TM3 (Ile172, Tyr176), TM6 (Phe335, Ser336, Phe341), TM8 (Ser438, Thr439) and TM10 (Thr497, Val501).”

We have further emphasized that support for our HDX signature from S-citalopram and cocaine binding also comes from MD simulations (line 334-336)

“Similarly, the cocaine binding site could be formed by Na⁺. This is supported by MD simulations showing that cocaine stabilized an outward-facing conformation, both in SERT²⁶ and DAT⁵⁰.”

As suggested, we have changed part of the discussion to be more descriptive and confirming earlier computational findings. We found this angle particular appropriate when discussing the ligands promoting the inward facing state, i.e. K⁺, 5-HT and ibogaine. The relevant part of the Discussion now reads (lines 360-364)

“Other significant differences in the stability of hSERT varied between the three states. In particular, 5-HT still causes stabilization of TM1a (residues 89-95, (Fig. 5)). This could suggest that the transition from outward-to-inward has yet to be completed. MD simulations have supported this suggestion by elegantly showing that the solvent accessible surface area on the SERT intracellular side was higher in the noribogaine-bound transporter relative to 5-HT bound²⁶.”

Finally, the Discussion has now also been considerably shortened as requested.

Reviewers' Comments:

Reviewer #1:

Remarks to the Author:

The revised manuscript has improved a lot.

Supplementary figure 1a should indicate by label that lanes 7-10 are BSA calibrations, and a PAGE-SDS gel of the SEC-purified SERT (load and purified) should be added to qualify the interpretation of suppl. figure 2

Although the latest version is not available to me (empty file) I assume from the "response to reviewer" that it has clarified the consistency with earlier findings and shortened in the discussion as requested.

REVIEWER 1

Comment #1:

The revised manuscript has improved a lot. Supplementary figure 1a should indicate by label that lanes 7-10 are BSA calibrations, and a PAGE-SDS gel of the SEC-purified SERT (load and purified) should be added to qualify the interpretation of suppl. figure 2

Our response: We thank the reviewer for the complement of our revised manuscript. We have now revised Supplementary Fig. 1a as requested. We have further added to Supplementary Fig. 2, two images of SDS-PAGE gels showing the purity of the hSERT sample prior to loading on SEC and the purity of the hSERT sample after SEC purification.